# Automatic Fault Detection and Diagnosis in Cellular Networks and Beyond 5G: Intelligent Network Management

**Arun Kumar Sangaiah** [1], **Samira Rezaei** [2], **Amir Javadpour** [3,*], **Farimasadat Miri** [4], **Weizhe Zhang** [3] **and Desheng Wang** [3]

1   International Graduate Institute of AI, National Yunlin University of Science and Technology, Douliu 64002, Taiwan
2   Leiden Institute of Advanced Computer Science, University of Leiden, 2311 EZ Leiden, The Netherlands
3   Department of Computer Science and Technology (Cyberspace Security), Harbin Institute of Technology, Shenzhen 150001, China
4   Computer Science Department, Ontario Tech University (UOIT), 2000 Simcoe St N, Oshawa, ON L1G 0C5, Canada
*   Correspondence: a.javadpour87@gmail.com

**Abstract:** Handling faults in a running cellular network can impair the performance and dissatisfy the end users. It is important to design an automatic self-healing procedure to not only detect the active faults, but also to diagnosis them automatically. Although fault detection has been well studied in the literature, fewer studies have targeted the more complicated task of diagnosing. Our presented method aims to tackle fault detection and diagnosis using two sets of data collected by the network: performance support system data and drive test data. Although performance support system data is collected automatically by the network, drive test data are collected manually in three mode call scenarios: short, long and idle. The short call can identify faults in a call setup, the long call is designed to identify handover failures and call interruption, and, finally, the idle mode is designed to understand the characteristics of the standard signal in the network. We have applied unsupervised learning, along with various classified algorithms, on performance support system data. Congestion and failures in TCH assignments are a few examples of the detected and diagnosed faults with our method. In addition, we present a framework to identify the need for handovers. The Silhouette coefficient is used to evaluate the quality of the unsupervised learning approach. We achieved an accuracy of 96.86% with the dynamic neural network method.

**Keywords:** cellular network management; network optimization; cellular network troubleshooting; data mining; unsupervised learning; supervised learning; self-healing networks

## 1. Introduction

Cellular networks' future is in heterogeneous networks. These networks consist of different radio and architecture technologies. On the other hand, the growing services and technologies increase the complexity of these networks and the internet of things (IoT) daily. For operators' survival in today's competitive world, it is necessary to use a new strategy to manage these networks. Using self-organizing networks (SONs) is the most common solution to achieve this goal. The architecture of SONs consists of three general parts: self-configuration, self-healing, and self-optimization [1,2].

Self-regulating networks include different phases, including designing, updating, or developing the network. Transmitter stations, base receivers, nodes, and other network terminals need to set related parameters, especially with the emergence of LTE networks based on IP and with flat architecture [3,4].

With the increase in the complexity of cellular networks, self-healing networks have become more popular. However, the rate of published studies and articles in this field is much more limited than in other areas. These networks can manage faults and system

failures by automated detection and identify their causes. Finally, the self-optimizing network calculates and values the network parameters that need to be reset, according to the network status, traffic, and services used by the users [5].

Therefore, actions related to self-regulating networks are in the development phase, or any activities related to resetting the network are in the operational stage. The self-optimization starts working in cases where there is no apparent problem in the network, but the cell performance is average. Finally, the measures related to the network's self-healing capability are used when the quality and efficiency of the network severely decline. Our focus in this paper is on self-healing networks. This research seeks to find some solutions to improve the process that is currently used in the industry to detect faults and identify the cause of faults. Therefore, all assumptions and modeling have been made according to the needs of the sector and consultation with experts in this field. In addition, according to experts in this field, it improved the quality of operational work.

### 1.1. Innovation, Importance, and the Value of Research

Considering the importance of cellular networks in everyday life and the fact that they are becoming an inseparable part of modern lives, paying attention to the faults of the network, trying to reduce the time a fault affects the network through early detection of the fault, and, also, the correct identification of the fault caused in the network so as to expedite a return to the network's normal mode are of particular importance.

The difference between this work and other similar works in this field includes the use of accurate data for modeling the behavior of the network in the face of various types of faults, and examination of all the characteristics and critical statistical indicators that optimization engineers use in the operators to analyze the faults that occur in the network. All features are reviewed simultaneously. Therefore, the effect these indicators have on each other, and the impact of combining the values of different indicators on the network's performance, are studied in this research. In addition, unlike most of the proposed methods, the construction of the proposed model in this field was achieved without the intervention of human resources and experts. Since there is a possibility of human fault in setting model parameters and the assigned values are highly dependent on the expert's experience, the fact that different experts may value the parameters in different ways, under the same circumstances, is pertinent. For this reason, the use of a method that rejects human intervention is significant. Finally, this research used three available sources of information in troubleshooting and optimizing the networks, which, according to our study, is the first case of research considering all sources. In other words, all related work in this field composed our first source of information, namely, the performance support system data. In the second section, these data are introduced in detail.

### 1.2. Global System for Mobile Communications (GSM) Architecture

The GSM network is a digital cellular communication system that operates by the cellular making of a geographic area and reusing frequencies for non-contiguous cells. This network consists of some components, such as the network and switching subsystem (NSS), base station subsystem (BSS), and operation support system (OSS). Figure 1 shows the GSM network architecture [6].

As seen above, the base station subsystem includes all the related radio capabilities of the GSM network and is responsible for establishing communication between the network subsystem and network users. Therefore, the BSS is composed of several base station controllers that manage the operations at base transceiver stations (BTSs) through the Abis interface. Each transmitter and receiver of the base station is responsible for serving the users of its coverage area through the A (air) interface [7,8].

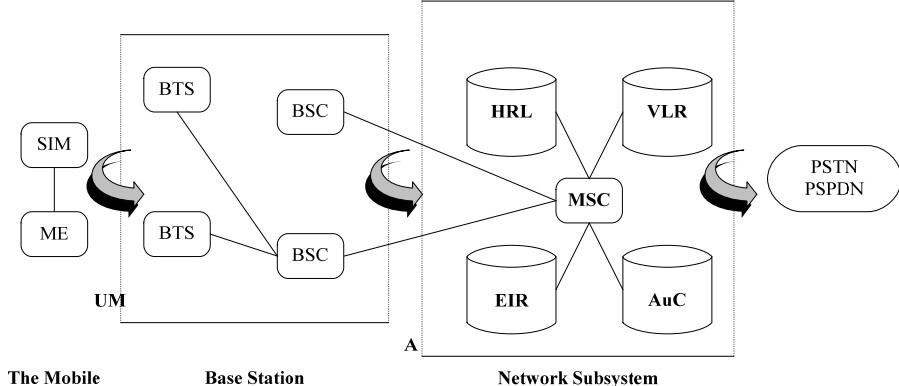

**Figure 1.** The architecture of the GSM network.

There are three types of data sources to analyze and check the quality of these networks and identify and troubleshoot the faults that occur in the networks. The most critical data used in the fault management of these networks is the statistical data reported from the BTS side, which indicates the status of various statistical indicators from different traffic and signaling perspectives. The second category of used data results from an experiment in the areas covered by the networks through human power. This test is a method to measure and check the coverage status, capability, and quality of the networks. This test is conducted by making a call with a mobile phone that can record the reported measurements and parameters from the BTS side. The name of this test in the industry is the driving test. In the third section, some detailed information about this category is given. Finally, the last category of used data in improving the network performance is the use of collected information from subscriber complaints. Section four reviews the registered reports of the Mobile Telecommunication Company of Iran (Hamrahe Avval).

In general, our main goals in this research are as follows:

- To use accurate data to identify the cause of a fault that occurred in the network (the required data was obtained in cooperation with the mobile telecommunication company, Hamrahe Avval.)
- To appropriately use available data sources to improve the accuracy of fault diagnosis systems
- To analyze different indicators' values and how they affect the occurrence of faults
- To model events and unpleasant events (faults) in the network
- To have a comprehensive view of the network and the fault types that occur in the network
- To provide a model that has a minor dependency on expert opinion.

### 1.3. An Introduction to Decision Support Systems

Decision support systems, as is evident from their name, are decision support methodologies. A DSS requires a knowledge base, a user interface, and a mechanism for processing the data in the database. A DSS is an information system that facilitates the decision-making process for managers using an interactive user interface. DSS uses online analytical processing (OLAP). Therefore, it requires analyzing a large amount of data related to the organization's past [9].

This project deals with the organization of different components of a DSS system that can correctly detect a fault and identify the cause of the obtained fault. Therefore, one of this project's goals is to provide a knowledge base and a mechanism for processing faults in the system.

### 1.4. Problem Definition

There are some significant problems with the existing methods for self-healing networks. First, most of the studies have used a simulator to model the system, and the

dynamics characteristic of radio networks are the most apparent features of this type of network. Therefore, if a simulator is used, the events that happen for these networks may be ignored in reality. On the other hand, as mentioned before, the number of published articles in the field of self-healing networks is less than in other areas. Thus, the researchers have not paid enough attention to one of the most critical issues in the field of cellular networks, due to the competitive ethos among operators and the need to keep customers satisfied in this industry. On the other hand, according to our studies, all conducted studies in this field are limited to using performance support system data that include key performance indicators. In this research, we tried to use other available resources to improve the accuracy of the presented model for fault detection and to identify the cause of the faults in the system [10].

### 1.5. Self-Healing Networks

As mentioned, the number of published articles in self-healing networks is less than in other fields. There are several main reasons for this. First, it is difficult to specify the common faults and the corresponding causes in these networks, due to the network dynamics and their dependence on the behavior and services used by the users. When different experts face a specific fault, they may have different opinions. On the other hand, there is no valid document in the literature in this field to correlate the fault and the cause of the fault in the network. The second reason is the complexity and difficulty of simulating faults and modeling the network mode in simulators. In addition, it is almost impossible to have an acceptable quantity of faults during the lifetime of a network. Due to the reasons mentioned, less investigative research has been conducted in this area than in other areas.

As was mentioned before, this research aimed to solve various challenges in the field of cellular networks' fault detection, to categorize different types of faults according to varying values of performance evaluation indicators, and to investigate the impact of each fault on these indicators. In other words, our research was conducted in the framework of self-healing networks. In general, the nature of a cellular network's components and systems is not immune to faults and failures. In the old methods, faults were detected by activating alarms. When remote methods could not solve the faults, radio network engineers were sent to the site to identify the fault. As expected, this process is very time consuming, and sometimes it takes more than a week for the system to return to its normal status [10].

On the other hand, some problems cannot be traced through alarms and are only identified if a complaint is reported to the operator by the customer. The future of these systems is improving with the use of self-healing methods. In general, self-healing methods address the process of remote fault detection, identify the cause of the fault, and, finally, set up the fault compensation system, or repair activities, to minimize the effects of the fault on the equipment and network components. For this purpose, in this section, we first present the available views and methods in the fault detection field and in the diagnosis of causes of faults in cellular networks. The logical flow of self-healing systems is given in [10]. This flow consists of three logical parts: identifying the fault, analyzing the cause of the fault, and, finally, setting up the fault compensation system. In the following, we briefly review the studies conducted, according to the categories presented in this study.

### 1.5.1. Fault Detection

This section is responsible for identifying faulty cells. Self-healing mechanisms continue to function only on these cells. Identifying defective cells is the most superficial aspect of the work, which may be assessed using alarms and indicator values. If the efficiency graph of a cell has a steep decreasing slope, that cell is considered disabled. Establishing a threshold for index values is a straightforward, but typical, method in this area [11]. For instance, a cell is recognized as faulty if the value of an index is less than (or greater than) the predetermined threshold. Other approaches, such as in [12], consider a profile for each index, which depicts the indicator's normal state. The issues with employing a thresh-

old are avoided since the fault detection stage continuously measures each indicator's divergence from its profile. A solution is offered in [13] to issues like personnel needing to correctly configure the parameters for profile building and deviation detection [14]. The Kolmogorov–Smirnov test compares the index's distribution with its ideal state as the functional approach. Each index's profile contains a record of the statistical distribution of the index in normal mode [14].

One of the current issues is that alarms may not detect all network faults. As a result, the network problem might not be discovered for several hours or days. Finding cells that are referred to as sleeping cells, and which are unable to send an alert to the center, is the key issue in this subject. On the other hand, it is impossible to guarantee the presence or absence of faults in the system by looking at the value of just one index. So, it is essential to assess multiple indicators' values [13], simultaneously.

It is possible to implement an intelligent and dynamic fault detection mechanism. According to the study in [15], defective cells can be divided into three categories: deteriorated cells that are still functional while having subpar quality, paralyzed cells, in which a severe fault occurred, but the fault has little to no negative impact on the user experience, and does not result in disgruntled customers, and network cells that eventually shut down and cannot carry traffic.

In [16], the process of detecting disabled cells is carried out by creating observation graphs, binary classification, and a list of nearby cells. Along with classifier quality, accuracy in recognizing the frequency pattern of network issues should also be considered. It was demonstrated that, despite the classifiers' high sensitivity in diagnosis, they produce many false alarms, making them unsuitable for industrial use.

Network faults have been found in [7] utilizing the Self-Organization Map (SOM) and K-means clustering methods. SOM is not only used in the scenario mentioned above. By creating the techniques for cell categorization, the researcher in [17] improved the control of the state of the cells in the cell network. An anomaly is defined as a significant data disagreement with SOM nodes. The use of a local threshold to identify abnormalities distinguishes this study from similar ones, because local approaches improve diagnosis accuracy and decrease the production of false alarms. The distribution of deviations in the network can be used with the adaptive threshold in this investigation [18].

Garc sets, categorization, and SOM are three data mining techniques that have been used to measure the performance of mobile networks [19]. These techniques are used to discretize the value of the indicators and to categorize the status of a network into good, normal, bad, and unacceptable network states.

In [20], data vectors representing the system's normal state were employed to perform the learning step of a neural network used to construct the normal state profile. The confidence interval for the normal/non-normal state was established through experimental analysis. To determine the reason for the fault, inference rules were produced using the trained network.

One of the tasks that may be mentioned in the context of fault detection is using the alarm correlation technique and delivering fewer alarms to operators. These algorithms are sensitive to minimum support and confidence levels for pattern identification methods. The method finds a high number of patterns when these parameters are set to small values and a low number of patterns when these parameters are set to large values. Since this research did not investigate the correlation of alarms, other pertinent details were avoided [7].

It should be emphasized that after recognizing the fault, factors, including its duration and the effect it has on the operation of the system, should be examined. If the observed fault fits the prerequisites, the process of determining its source and compensating for it begins.

### 1.5.2. Identifying the Cause of Fault

– This section suggests remedies for a cell's aberrant and dysfunctional condition. As a result, it is crucial to start by limiting the fault within a cell. The nearby cells should

work together to reduce the fault's detrimental impact if it cannot be swiftly fixed. Corrective actions must be conducted to eliminate the fault when the fault's root cause is identified. The following section goes into more depth about this phase.

– Hardware and software issues may be causes of cell faults, and other potential causes include incorrect configuration, inadequate coverage, and interference. As previously stated, alarms alone cannot identify the source of an issue and additional data, such as the values of the indicators, must also be examined.

– The following categories serve as a general summary of the current methods used in this field. Here, the primary focus is on techniques based on data mining and finding significant patterns in the data to identify the cause of the fault.

– Methods based on data mining

– Statistical methods

Data mining techniques are among the approaches adopted in the automation of network fault detection, due to the enormous and growing volume of data accessible for the evaluation of the quality of cellular networks. The absence of a database of classified instances, highlighted in the introduction section, is one of the most significant issues in this sector. Moreover, this area is separated from other areas related to identifying the cause of faults, due to the continuous nature of the performance evaluation indicators and the existence of logical faults that are not dependent on the physical parts of the equipment and are probably related to the wrong settings of the equipment. For instance, Rule-Based Systems (RBSs), which are a subset of Expert Systems employing a set of "if–then" rules [10], did not gain enough success in cellular communication networks because they did not perform successfully in cases with both large and unpredictable ranges. Making rules in dynamic and unpredictable contexts has its own set of issues.

The two main divisions of data mining methods are unsupervised and supervised methods. The following sections review the methods used in these two types.

1.5.3. Methods Based on Data Mining

• **Supervised Methods**

Due to the issues that exist in this domain, and have already been referred to, the present methods in the area of supervised methods primarily investigate in a restricted way. The techniques employed are often limited to Bayesian networks. The most effective techniques in this area are discussed in the following sections.

There has been much interest in using Bayesian networks to represent network events and communicate between faults and their causes [21]. The primary distinction between these approaches depends on the structure used for identifying the cause of the fault and the various fault indicators and reasons that the suggested solution can look into. A model for locating defects in cellular networks using the Naive Bayesian classification method was introduced in [21]. The fault causes are treated as classes, and the values of various indicators are treated as features. The introduced model, providing the knowledge base used for classification, is divided into two primary categories: quantitative and qualitative. The quantitative section comprises the links between these components, and the qualitative part includes the model's components. This article demonstrated how the average beta density function parameters could be used to thoroughly specify the relationships between the model's parts. By discretizing the index values and using the Spartan Bayesian classification approach, [22] was able to identify the fault source. The entropy minimization discretization (EMD) method was applied to boost the effectiveness of the proposed model. Utilizing this technique resulted in the best possible selection of useful indications from the input parameters.

The study in [23] aimed to improve a network's accuracy by considering the continuous nature of the index values. Giving them discrete values decreased the precision of the results produced. As a result, the Smooth Bayesian network was employed with varying state uncertainty. On the other hand, as these parameter values were considered, more data was required to calculate the parameter values. The findings of the applied method,

which used Bayesian law, are displayed in [24], utilizing a dynamic simulator system on the UMTS network. The architecture for employing these networks' troubleshooting tools was provided. It is always necessary to strike a balance between the precision and complexity of a model. The DCR index calculated the number of disconnected calls, which was the fault detection criterion in [20] (thoroughly explained in the second section). The system experienced a fault if the value of this index exceeded the threshold, and the causes of the fault were determined by looking at the simplified Bayesian network and utilizing the Independence of causal impact (ICI) approach between the indicators and the faults' causes. The simplified Bayesian technique was dealt with using the ICI method, and each fault source was employed as an independent node with two true or false states in the modeling. The outcomes of the simulations demonstrated that the effectiveness of these two approaches was equivalent but that the ICI approach was more complex. As a result, the simplified Bayesian model was given precedence. These techniques employed reverse engineering to create faults under particular circumstances. In other words, the methodologies utilized in the paper attempted to model the fault using the value of the key performance indicators when the fault occurred, even though the source of the issue was previously known.

The absence of a database of classified instances that can be utilized to estimate the parameters of the suggested model is one of the issues in this subject. A technique for developing a statistical model using expert knowledge was provided in [25]. In this method, knowledge acquisition was accomplished in two stages: first, with the assistance of experts, knowledge was gathered, and then models were built. Information about the various types of faults, variables that influence the various types of faults, the relationship between the types of faults and the influencing variables, the threshold for each variable, the likelihood that each variable influences faults, and, finally, comparison of the information obtained were all related to the data collection stage. Finally, this method constructed the model via the Bayesian network approach. It was proved that when parameter values were calculated using statistical features, rather than expert judgment, systems to detect the causes of faults performed more accurately.

In [26] a strategy for locating the fault's root cause, that combined neural network and law-based techniques, was offered. The rules matrix and a hierarchical and distributed multi-dimensional structure were employed to gain the capacity for parallel reasoning. However, the author concentrated on the Internet network and did not estimate the accuracy of this method on cellular networks. Due to the neural network's high power in detecting data behavior, it was also employed in research in [27] in such a way that the criteria for determining the cause of a fault were applied to each type of fault that the neural network specified. These rules could use other available data to check network efficiency and deviation of neighboring cells to ensure that one did not affect the other.

- **Unsupervised Methods**

Among the unsupervised techniques, the Self-Organization Map (SOM) is the most popular one. This neural network-based technique is usually utilized for data presentation. This process transforms high-dimensional data into two- or three-dimensional data, preparing it for presentation [28]. According to [29], the SOM algorithm was given a predetermined set of features, and after clustering the data, each cluster was utilized to identify the various fault types, their sources, and the times when they occurred. This technique was only used in regard to the difficulties of signaling channel capacity and traffic flow analysis. The study's researcher in [29] began by using SOM to decrease the dimensions of the data to two or three dimensions. After applying clustering techniques, such as hierarchical approaches, and division methods, such as k-means, to the data, the clusters were then assessed using the Davies–Bouldin index.

1.5.4. Statistical Methods

Identifying the cause of faults in [30] was done by comparing the values of numerous indicators and taking appropriate corrective action, based on expert judgment. The opera-

tional approach involved ranking the causes of the set of indicators that strayed from their profile in decreasing relevancy. The scoring system employed in this approach determined the likelihood of a specific cause for an occurred fault, using the set of indicators that deviated from the baseline.

In [31], the network status was examined using the characteristic vector of the network status, which was as long as the number of the analyzed indicators. This characteristic vector considered the threshold limit for the six most significant indications. The Hamming distance was used to determine how various records differed. As a result, the fault cause was the same for records that belonged to the same category. The expert specified the number of categories.

A multilayer hierarchical structure was used in [32,33] for fault-cause couples. The collection of current criteria was classified according to the knowledge of industry professionals. Counters were utilized in this study as an alternative for key performance indicators.

### 1.6. Compensating the Occurred Fault

The purpose of this section is to suggest relevant action to solve problems. In [10], these measures were divided into simple and parametric. Simple activities are those that can be executed without additional arguments. In other words, there is a direct relationship between the cause of the fault and the necessary measures to fix it. For example, if the cause of the fault is, in part, a hardware problem, the corresponding action is to send a technician to the site and replace that part. On the other hand, parametric measures are not easily determined and require the implementation of different algorithms. For example, if it is necessary to change the parameter value, its new value should be calculated. Since the number of parametric measures is large, it would be valuable to introduce an algorithm to identify the standards. To compensate for the fault in the study in [34], changing and resetting the parameters of neighboring cells, and optimizing the covered area by them, was utilized. Things such as filling the space of the disabled cell, resetting the antennas' power consumption, and choosing the correct value of the parameters through fuzzy logic and reinforcement learning were performed in this study. In general, the necessary measures in correcting the occurred fault depend entirely on the detected fault in the first step and the cause of the detected fault in the second step. Among the measures to fix the fault are resetting the parameters, reducing the angle of the antenna, or increasing the strength of the received signal by the user.

The distinctive feature of all past works is the limited view of the faults in the network, because, by considering this field's data complexity and the uncertainty in the number of parameters assigned by experts for a model's relationships, it was difficult to consider a large number of network events. On the other hand, the accuracy of the obtained model from this method strongly depends on the expert's knowledge and the assigned value to each parameter. Thus, this article focuses on designing a model that can perform the initial stages of model construction with minimal dependence on expert opinion and only using available statistical information. In addition to reducing the model's dependency on humans, the accuracy of the produced model is increased by removing the effects of human error. The general framework of the conducted work of this article is illustrated in the following figure. As seen in Figure 2, this paper uses three types of data: performance support system data, drive test data, and customer contact center data. This paper's second and third sections discuss the first two sources of information. These three sources of information on the network mode can determine the existence of a problem in an area. In addition, using the information fusion methods, this system can automatically identify the fault causes in cellular networks with the help of three different data types.

The structure of this article is as follows. In the second section, after getting familiar with the indicators used in the qualitative assessment of the network condition and the types of faults affecting the value of each indicator, the essential source of available data, called the performance support system, is examined in detail. The results are evaluated to analyze the cause of a fault. The third section introduces another source of available

fault diagnosis: drive test data. In the fourth section, after submitting the third source of data, subscriber complaints, the results of the previous two sections are reviewed and examined. Finally, in the fifth section, we summarize, conclude and provide a perspective for future work.

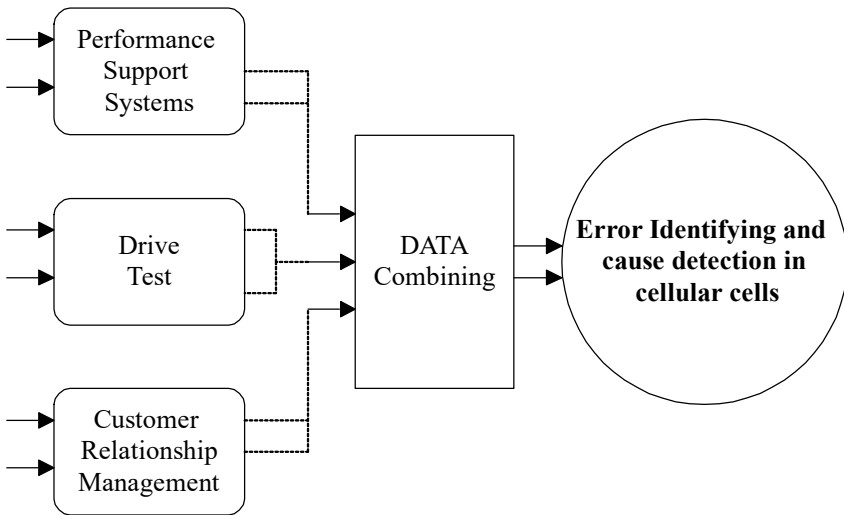

**Figure 2.** An overall framework of the conducted works in this paper.

## 2. Performance Support System

As mentioned in the previous section, one of the used data sources in troubleshooting cellular networks is the available data in the performance support system. This system collects statistical data reported from transmitter and receiver base stations. This data examines the network mode from various aspects. Key indicators are calculated using counters that record all the events that have occurred in the network. These key indicators, comprehensively explained below, are evaluated to troubleshoot in the performance management part of cellular networks. The values of these indicators determine the cause of the fault in the system, and help industry experts improve the system's performance. The data we used was supported by the Mobile Telecommunications Center (Hamrahe-Avval). These data were related to Tehran's eighth district and included 50 primary transmitter and receiver stations.

In the following, we first make a preliminary statement about the channels used in the cellular network and the process of making calls in these networks. We examine the measured indicators in the performance support system in detail, and, finally, we describe the applied method in regard to these data.

### 2.1. Definitions, Principles, and Theoretical Foundations

The first step in troubleshooting cellular networks is to identify faulty cells. A problem in the network refers to a situation that hurts the quality of network service. Different operators use different methods to identify issues in the network. It is first necessary to know the cellular network and the key performance indicators in troubleshooting in the network to identify its network faults.

#### 2.1.1. Standalone Dedicated Control Channel (SDCCH) Signaling Channel

This channel is dedicated to subscriber signaling activities. Among these activities, we can mention the basic steps to making a call, updating the physical location of subscribers, and sending and receiving short messages. Each subscriber occupies a capacity of the signaling channel before making a call in the network.

### 2.1.2. Traffic Channel (TCH)

This channel, accounting for about 75% of the radio resource range, is used to transmit sound in the network. The network signaling channel provides about 25% of the network resource capacity.

### 2.1.3. Key Performance Indicator (KPI)

The key performance indicators in the cellular network management literature are measures that can be used to measure the quality of a network at different levels. Some of these indicators depend on time or user behavior, such as the number of downlinks to the user. Other indicators, such as the channel quality indicator (CQI), random channel access attempts (RCATs), or the percentage of dropped calls, due to lack of dependency on the user program used on the user side, are less related to the user's behavior, so the indicators of this category are more suitable for evaluating the network condition. Telecommunication networks are changeable, and there are several key performance indicators to assess network condition. Using manual methods to solve network problems does not meet the management needs of these networks.

Many indicators are used to evaluate the quality of radio networks, and by saving the fault-free behavior of each KPI, its deviation from its normal behavior can be identified. The high number of these indicators, and the related counters to each one, have made it impossible to check all the indicators and the impact of each one on the quality of the network. In the following, we examine the steps of establishing a call in a cell network. Then we introduce the most important indicators affecting the network mode, submitted by experts in this field.

### 2.1.4. Procedures for Making a Call

From the moment the subscriber initiates a call to when the traffic channel is successfully assigned to him or her, various steps are performed in the network. In Figure 3, the different stages are illustrated, along with the indicators used to measure the statistical status of the network. These indicators are fully explained in the following part of this section.

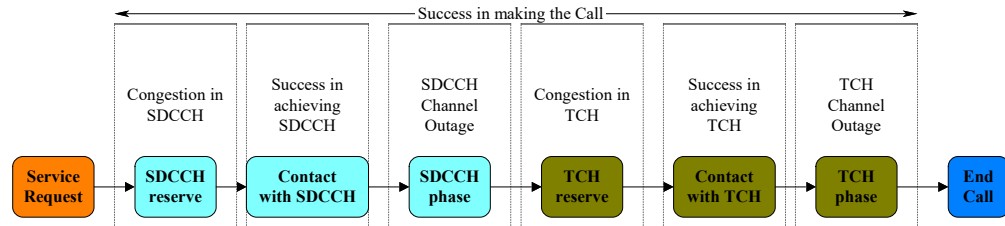

**Figure 3.** The steps of making a call in the GSM network with the network performance measurement indicators.

### 2.1.5. Introducing the Most Important Key Performance Indicators

As seen in the above figure, several indicators measure the network quality from different aspects. For this purpose, before identifying faults in a network, it is first necessary to thoroughly examine these indicators and the factors that affect them.

1. **SDCCH Congestion**

Traffic channels in radio networks carry the main network load, while control and signaling channels are used to achieve this goal. These channels are used to control the behavior of mobile phones, to communicate, to control established calls, to facilitate call handover, etc. If subscribers' demands exceed the available resources, congestion occurs in this channel. One solution to this problem is adding more resources and physical capacity to the network resources.

2. **SDCCH establishes success rate**

This indicator shows the percentage of successful requests to take over SDCCH. One of the reasons for a negative value for this index is downlink interference, which occurs when there are too many subscribers' location updates, due to network design issues, and hardware issues.

3. **SDCCH drop**

When a network user makes a request to communicate with another subscriber, or when a network user is called by another network subscriber (through an incoming call), there is a need to take over the SDCCH. Some reasons, such as the timing advanced (TA) timer, low strength of the downlink or uplink signal or both, or low quality of the downlink or uplink signal, affect the SDCCH drop. In addition, if the traffic channel is congested, the requested call is returned on the SDCCH.

4. **SDCCH means holding time**

This index indicates the required capability of the SDCCH for high-speed access. That is, it measures the mean time the channel is held due to switching between services. The longer this time, the worse the service provided.

5. **TCH Congestion**

Congestion in the network means insufficient resources are available to make calls on the network. Congestion in the traffic channel is one of the most critical problems of the radio network, and it strongly affects the quality of the network. Congestion leads to higher subscriber dwell time in the SDCCH. One of the ways to reduce congestion in the network is to use more radio resources. Unfortunately, there are many limitations in terms of economy, lack of space for installing resources, etc. Therefore, it is recommended to use other methods, such as using the "half rate" technique. In this method, each period is assigned to two subscribers instead of serving one subscriber. Although the use of half-rate improves the network congestion situation, it reduces the quality of the received signal by the user. Another way to deal with congestion is to share the traffic load of the cell and transfer calls to neighboring cells. Like the half-rate technique, this method has the problem of reducing service quality.

6. **TCH Assignment Success Rate**

This indicator directly affects the user experience and shows the TCH assignment success rate percentage. If there is congestion in the traffic channel, the value of this index decreases. Other factors, such as the amount of coverage, interference, and hardware problems, also reduce the percentage of this index.

7. **Call Setup Success Rate**

This indicator, one of the most important key performance indicators used by all mobile phone operators, indicates the successfully launched call rate. There is no single standard method for measuring this index; therefore, every operator can calculate it differently. High values of this attribute indicate good network performance because a high percentage of calls are successfully established. Many reasons can be listed for inappropriate values of this index, including weakness in signal strength, congestion in the traffic and signaling channels, disconnection in the SDCCH, and failure in the appointment of the TCH.

8. **Call drop Rate**

Once the user has successfully taken over the traffic channel, due to various reasons, he or she may lose connection with the network during the call. The most important reasons that lead to the interruption of a call within a network are weak power and poor signal quality. Improper coverage of the radio path, hardware faults, and non-optimal setting of parameters affect the signal strength. Interference, issues related to coverage and handover

(neighbor loss, incorrect settings in positioning algorithms), and, finally, the wrong set of parameters, can also cause deterioration of the quality of the signal received by the user.

9. **Handover**

The handover process is conducted by controlling positioning algorithms. Each BSC measures some information, such as the downlink signal strength and quality of the mobile user and the downlink signal strength and quality of the user's six best neighbors, when making a handover decision. The "Handover failure Rate" index provides valuable information, such as the efficiency between two neighbors, the strength or weakness of the typical radio path between two neighbors, issues related to interference at the border between two neighbors, etc. In addition, to identifying weak neighbors in the radio path, handover optimization is used to improve the call failure percentage due to handover failure. Therefore, the incorrect design of neighborhoods, traffic congestion at the destination, and the fault of swapped sectors in implementing antennas lead to handover failure.

10. **UL/DL (UpLink/DownLink) Signal Quality**

This indicator, which has been introduced in the previous sections as one of the reasons for success or failure in the most important network quality assurance indicators, calculates the quality of the sent signal through the base transmitter and receiver station to the user (the downlink signal), as well as the sent signal from the user to the station (the uplink signal). If the received signal strength by the user is less than the amount specified in the existing settings of each operator, the user's call is disconnected. In addition to signal quality affecting the increase or decrease of the call disconnection rate, other factors. such as access to the traffic channel and signaling interrupted in the signaling channel, etc. can also play a part.

11. **The amount of half-rate capacity usage**

There is no such measurement indicator in the base transmitter and receiver station. Still, in the transmitter and receiver station, the network's transmission traffic is measured at the total and half rate. The half-rate capacity is the high number of call requests in an area, which cannot increase the power in that area. Therefore, they reduce the call quality in such areas by converting the full-rate channel to half-rate to cover more calls and support more calls. This article uses this feature to identify the amount of network load.

For each KPI, it should be determined whether it functions similarly to its profile or whether it has a specific statistical change compared to it. One of the available ways to check this situation is to define a threshold on the KPI values and divide the range of their values into two categories: above the threshold and below the threshold. More advanced techniques for fault detection involve checking whether each KPI passes the threshold value continuously for a specific time or passes the threshold value a certain number of times. All threshold-based methods face the problem of quantizing the range of KPI values into a binary space below the threshold and above the threshold. However, this means other helpful information is lost, and it is impossible to reliably detect faults from these data. In the following, the architecture of the proposed method, in regard to these data, is explained.

*2.2. Architecture of the Proposed Method*

To achieve the goal of this research, which was to detect a fault and identify its cause, we used the following framework. In the next part, we explain this architecture's different aspects in detail (Figure 4). The work process entailed preparing the available data, which was the raw data received from a BSC in Tehran, for the next operation. To ensure data validity, it was necessary to consider only the high traffic hours of the day. Since the data were, in fact, statistical characteristics of the network condition, they were reliable and valid only if the frequency of the used network was acceptable [10].

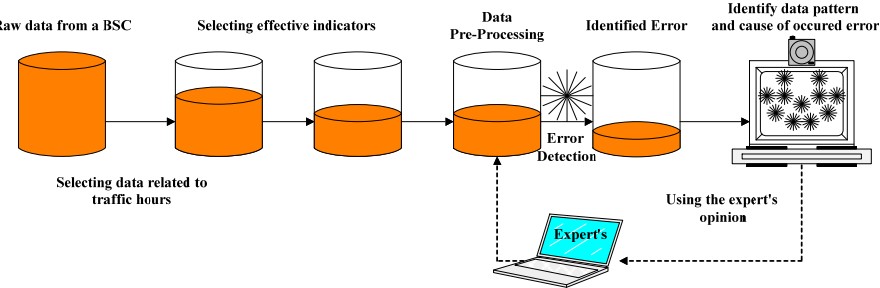

**Figure 4.** Architecture of the proposed method with performance support system data.

Among dozens of measured characteristics in network equipment, only some indicators, having the most significant impact on achieving the goal, were considered. Then, data preprocessing and cleaning operations were performed. After that, for fault detection in the system, a criterion for evaluating the quality of the network was introduced to detect the unfavorable network mode by examination of this value. Identified faults were entered into the next stage of the model to identify the cause of a fault. We tried to determine the cause of a fault that occurred in the network by using clustering algorithms. Finally, the obtained results were shared with an expert for evaluation. According to the expert's opinion, any necessary changes were applied to the model, and the process was repeated [35].

*2.3. Feature Selection*

After selecting the valid data, the first step to working with the data was to choose the effective indicators and features to achieve the goals. There are two ways to do this. One makes use of feature selection algorithms, and the other uses expert knowledge to select the most practical features. Since our available data was not marked to identify the cause of faults in the network, the only known solution was to use unsupervised algorithms and methods. Due to the problems in the field of feature selection in unsupervised methods, expert knowledge was used to select compelling features. In addition, since the foundation of this research was based on the industry and experts' opinions, trusting the experts' views, and following the industry's applied policies, could help us achieve the goals of this research. Therefore, the selection of features in this research was based on expert opinion.

*2.4. Data Preprocessing*

Since our method was based on data and all the results of our work were the result of working on the data, it was necessary to prepare the data for operations to increase the method's accuracy. The management of the missing values and the identification and removal of outliers were among the actions taken to clean the data before performing other operations on the data. Thus, all records containing missing values and outliers were removed from the data to simplify the work.

In addition, because the value of the analyzed indicator occurred in different intervals, it was necessary to replicate the index interval before applying any operation to the data. For this purpose, we used the minimum–maximum normalization method. Thus, the set of values of each index would be in the range of zero and one. To normalize each index with the minimum–maximum plan, after obtaining the minimum and maximum values of each index, the value of the index in each record was subtracted from the minimum value, and the result divided by the effect of removing the maximum values from the minimum, which indicated the data range. The following relationship shows this method [4,36,37].

Equation (1): normalizing data by the minimum–maximum method:

$$Normalized\ Value = \frac{the\ value - min(values)}{data\ range} \tag{1}$$

*2.5. Fault Detection*

Cellular networks are not immune to faults due to equipment, weather conditions, and user behavior. Various faults reduce the quality of network service. An alarm is generated in the system for any abnormal situation in the network. The high volume of generated alerts in network management systems has dramatically reduced the possibility of using these alerts in a targeted and effective way. They are used only in unique and rare cases. The key performance indicators discussed in detail in the previous section were the basis of our work and, in fact, the origin of the work of the operators' performance management department. Since our method was completely designed based on the industry and operators and related industries' needs, and was completely practical, it was used to detect faults in terms of the standard used by Iranian operators.

2.5.1. Defining the Network Quality Assessment Criteria

Fault detection is the most straightforward step in self-healing systems and networks. The complete, correct call rate (CCC) was the method used in the first mobile operator for fault detection and was the basis of our work. Therefore, only one criterion was used to check the network status. As mentioned in the previous section, there are many indicators to measure the quality of the network in different sectors and parts, each of which examines the network from only one perspective and the statistical information provided by each index is based on one of the aspects of the network. The research team of Sofrehcom Company provided a measure using indicators and criteria to evaluate the network condition. This criterion was obtained as the product of the following indicators:

1. Congestion percentage in the signaling channel (SD-Cong)
2. Success in signaling channel adjustment (SDestab)
3. Certain percentage in the signaling channel (SDdrop)
4. Traffic channel congestion percentage (TCHcong)
5. Traffic channel assigning failure rate (TCHAssignFR)
6. Mean uplink and downlink signal quality (RxQual)
7. Disconnection in traffic channel (DCR)

Equation (2): Calculating the qualitative measure of network performance:

$$
\begin{aligned}
CCC = (1 - SDcong) * SDestab * (1 - SDdrop) * (1 - TCHcong) \\
* (1 - TCHAssignFR) * RxQuality * (1 - DCR)
\end{aligned}
\tag{2}
$$

As can be seen, the CCC index consisted of the product of the seven most important indices. Therefore, if only one index had an unfavorable situation, its value would affect the result and the value of the CCC index. Therefore, by examining the importance of only one index, the status of the most important indicators of the network could be acquired. The fault detection process was performed by checking the value and behavior of this index during one month of available data. In this process, a histogram diagram of the index's monthly behavior was drawn in the investigated area, and the knees of the graph were evaluated as characteristic points in the behavior of the index. This method was used to discretize the performance of an index in a period into general categories, such as good, normal, bad, and unacceptable in [38]. Our priority was to work on a set of records that were in worse condition. This meant that network performance management and optimization departments had to first check the network parts in a worse situation. After troubleshooting these parts, other parts of the network could be inspected and evaluated. Therefore, we focused on the worst aspects and cells of the network. Other analyses and reviews were conducted on this part, and we reviewed different parts of the network.

2.5.2. Indicators' Analysis

As discussed in the previous section, we considered parts of the network as having faults and unacceptable status using the CCC index. We investigated the parts of the network that needed quick reaction and action to improve the conditions. Therefore, our

data was divided into fault and non-fault categories. As a result, we could use classification algorithms, including a decision tree, to analyze the network status and check indicators. The questions we sought to answer in this section included the following [10]:

1. Which indicators play a more effective role in fault detection?
2. Which combination of values of indicators leads to faults in the system?
3. The value of which indicators have a higher impact on creating faults in the network?
4. In general, what are the indicators of network faults caused by violations?

### 2.6. Identifying the Fault Cause

Identifying the cause of the fault was the most crucial goal of this research. It was necessary to perform several actions on the data to achieve this goal. Therefore, at first, the list of effective indicators was selected, according to expert opinion, from dozens of measured indicators in operators. In the next step, the non-deterministic relationship between the causes of fault and the values of the considered indicators was checked. The non-deterministic relationship meant that, for a specific cause of fault, the value of an index might be different at different times and in other cells.

On the other hand, the variance of the values of an index for a specific fault caused at different times and in different cells might be significant. Therefore, it would be challenging to find an index that had an unusual behavior for a particular cause of fault [39]. The figure below shows the probability distribution of different index values in the face of other faults in the network. As can be seen in Figure 5, it was impossible to deduce the fault caused by only considering the index value.

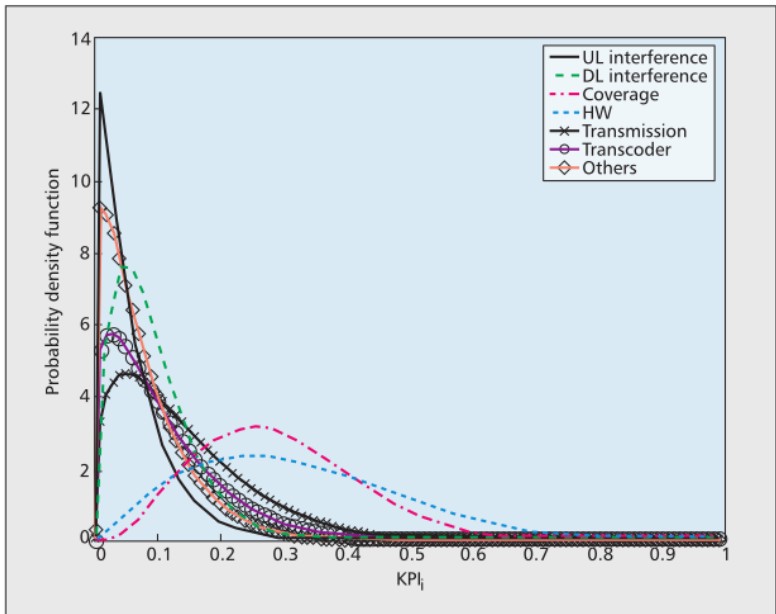

**Figure 5.** The possible distribution of different indicator values facing different faults.

In addition to identifying the necessary action to fix the fault, sometimes it is essential to consider the cost of doing so. Sometimes it is possible to do a replacement at a lower price. For example, if there is a hardware problem and the solution is to replace that part, it is better to restart it due to its low cost. Cost can include the cost of implemented time, the cost of executed action, its price on the network, etc. In the following, we review the steps to identify the fault caused.

### 2.6.1. Examining the Data Tendency to Clustering

The most important and main step in self-healing networks is identifying the cause of the fault. Since the available data were raw data, not labeled for the reason faults occurred,

it could be said that the information was unsuitable for classification methods. It was necessary to prepare the data for clustering methods.

Since the data of cellular networks have not been used in clustering methods in these dimensions, it was necessary to ensure the clustering capability of the data. For this purpose, the Hopkins coefficient was used. The Hopkins coefficient quantitatively indicates the data's tendency to cluster [40]. In this process, some random points are generated among other issues in the data and the distance between the random points and the fundamental points of the data is compared with the distance between two actual points in the data. In this way, it tries to ensure the non-randomness of the data pattern and the possibility of finding clusters that show the behavior of the data. The following relationship shows how to calculate this coefficient.

Equation (3): Hopkins' coefficient:

$$H = \frac{\sum_{i=1}^{n} q_i}{\sum_{i=1}^{n} q_i + \sum_{i=1}^{n} w_i} \tag{3}$$

where $q_i$ represents the distance between the generated random points and the nearest real data records, and $w_i$ represents the distance between the real records and the closest real neighbor in the data. In this way, the closer the value of the Hopkins coefficient to one, the more the data tends to be clustered.

### 2.6.2. Classification of Indicators

As mentioned before, it was necessary to select several indicators to perform operations on the data from dozens of indicators measured in managing cellular networks. Considering the characteristics of the cellular network, such as the dynamism and high variability of these networks, the simultaneous use of all these features would not lead to proper clustering of the data. On the other hand, since the steps of making a call and using the cellular network are two separate parts of receiving the signaling channel and receiving the traffic channel, these indicators have good separation capability. Therefore, the segmentation of the feature space was achieved according to expert opinion to increase the accuracy of the clustering algorithms and to obtain more favorable results from a practical point of view. Table 1 represents the list of indicators used according to the division.

**Table 1.** The list of indicators used in the traffic channel.

| Signaling Part | |
|---|---|
| The indicator's name | the Persian equivalent |
| SD-Congestion | SD channel congestion percentage |
| SD-Drop | Communication percentage on SD channel |
| SD-Estab | SD channel selection success rate |
| CDR | Communication percentage on TCH channel |
| TCH-Congestion | The percentage of congestion on the TCH channel |
| TCH-Assign-FR | TCH channel adoption failure percentage |
| RxQuality | Average uplink and downlink signal quality |

### 2.6.3. Data Clustering

The purpose of this step was to find meaningful clusters in the data. This meant trying to estimate the cause of faults in the system by finding meaningful patterns of network behavior in the data. Since any fault occurring in the network affects the value of one or more features simultaneously, the fault's cause could be estimated by considering the distribution of indicators in each cluster. The results obtained at this stage were provided to the expert to evaluate and interpret the distribution of data in each cluster, as well as to take a comprehensive look at the values of all indicators related to each category. In the

next section, we discuss the obtained results from experts in detail. So, our goal at this stage was to find clusters that could model the behavior of the data well.

Among the clustering methods, various methods were applied to the data. Among these methods, the Expectation-Maximization process provided a more suitable answer to our data. The details of the results obtained in evaluating different clustering methods are described in the next section. In the following, we explain the mechanism and working method of the EM algorithm.

The Expectation-Maximization algorithm, or EM method, is an iterative algorithm that estimates maximum likelihood. This algorithm starts with an initial estimate of θ and iteratively improves this initial estimate toward the observed data. This algorithm should be continued until the difference between the two estimates, 1 + t, and the estimate number, t, is less than the threshold limit (Algorithm 1). Otherwise, the convergence condition is satisfied, and the algorithm ends. In the following, we review the steps of this algorithm, where i represents the ring index, θ represents the estimate, and T represents the threshold limit for the algorithm to end [41].

---

**Algorithm 1** Expectation-Maximization method 1 begin initialize $\theta^0$, T, i = 0

---

Architecture of the proposed method with performance support system data:

Start
Raw data from a BSC
        Selecting effective indicators
        Data Pre-Processing
        Selecting data related to traffic hours
        Identified Fault
        Identify data pattern and cause of occurred fault

  begin initialize $\theta^0$, T, i = 0
          <u>do</u> i → i + 1
          E step: compute Q $(\theta \ ; \ \theta^i)$
          M step: $\theta^{i+1}$ → argmax Q $(\theta \ ; \ \theta^i)$
          <u>until</u> Q $(\theta^{i+1} \ ; \ \theta^i)$–Q $(\theta^I \ ; \ \theta^{i-1}) \leq$ T
  return $\hat{\theta} \ \theta^{i+1}$
  end

---

By experimenting with the data set, the detailed results of which are given in the next section, the EM method was selected from other clustering methods to categorize the available data. Due to the complexity and dispersion of data in this field, to achieve quality clusters, parts of the data that were not well clustered were re-clustered with the EM algorithm, and the results obtained were provided to the expert for final evaluation.

2.6.4. Evaluation of Produced Clusters

In this step, we evaluated the obtained clusters. Due to the lack of access to the gold data, we used intrinsic measures to evaluate the clustering results with the correct mode. Intrinsic measures assess the quality of obtained clusters without the need for external information. For example, the criteria that check the degree of adhesion or separability of the data in a cluster are among the internal criteria for evaluating clusters. On the other side are external measures. These criteria, which are supervised methods, are used when gold data are available, and the quality of clustering algorithms can be evaluated by comparing them with the target data [42]. In the following, we consider the internal methods that were compatible with the limitations of the work in this project and were used to evaluate the produced clusters. It should be noted that the same coefficients were used to obtain the optimal number of clusters, and the value of these coefficients was calculated for different numbers of clusters. The optimal value of these coefficients indicated the optimal number of clusters.

The Davies–Bouldin method is one of the most common ways of internally evaluating clusters obtained in clustering algorithms. The problem with this algorithm is that its excellent evaluation value does not use all the hidden information in the data and does not recover all the information [43]. The Silhouette coefficient, another internal evaluation measure of clustering methods [44], was used in this research. Both of these methods are based on the stickiness of the data in one cluster and the separability of the data with respect to other clusters. In the following, we review the details of these two methods.

✓ **Silhouette Coefficient Method**

The Silhouette coefficient is one of the most common ways of internally evaluating clusters obtained in clustering algorithms, and is based on calculating data adhesion and separability. Unlike the Davies–Bouldin method, which considers only the centers of the clusters, this method performs its calculations on all the points of the clusters. The general relationship below shows how to calculate the Silhouette coefficient.

Equation (4): The general formula for calculating the Silhouette coefficient:

$$S(o) = \frac{b(o) - a(o)}{max\{a(o), b(o)\}} \tag{4}$$

where *a(o)* indicates the cluster stickiness containing record o and is calculated through the following relationship.

Equation (5):

$$a(o) = \frac{\sum_{o' \in C_i, o' \neq o} dist(o, o')}{|C_i| - 1} \tag{5}$$

And

Equation (6):

$$b(o) = \min_{C_j : 1 \leq j \leq N} \left\{ \frac{\sum_{o' \in C_i} dis(o, o')}{|C_j|} \right. \tag{6}$$

where $dis(o, o')$ calculates the Euclidean distance between two points of o and o'. Therefore, *a(o)* calculates the average distance of point *o* with all the points that are in the same cluster with *o*. The value *b(o)* also calculates the average distance for all clusters from point *o* with all points that are not in the same cluster with *o*. Finally, for each cluster, the Silhouette coefficient value is obtained by averaging the values of $S_i$ in each cluster. The following figure shows an interpretation of Silhouette coefficient values for two clusters. In fact, the value of this coefficient is between −1 and 1. If $b_i$ which indicates the distance of each point from other points that are not in the same cluster, is smaller than $a_i$, which indicates the distance of points within a cluster, the result of the Silhouette coefficient is smaller than zero and has negative values. The closer the value of this coefficient to 1, the more correct is the assignment of each point to the cluster similar to itself. If the value of this coefficient is zero, it means that the point is located on the border between two clusters (Figure 6).

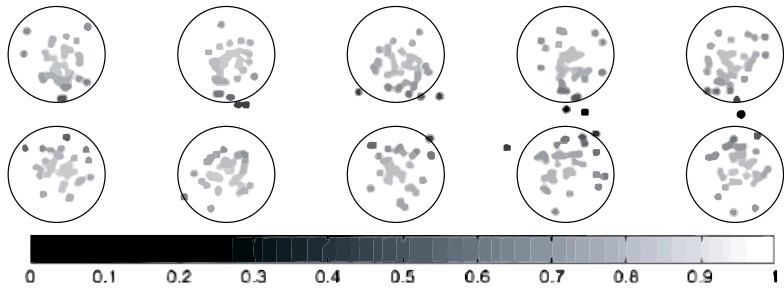

**Figure 6.** The Silhouette coefficient values' interpretation.

## 3. Results

In this part, we examine the obtained results from the data analysis of the performance support system. For this purpose, we first introduce the used data in this research and then report the obtained results.

### 3.1. Introduction of the Data Set

This data set, which included the most essential and central part of our work, contained statistical indicators of the quality mode of the network, which were collected by the transmitter and receiver base stations in real time. These data were averaged every hour and sent to the operators. The available data for conducting this research was the result of an hourly average of the events and the statistical status of the network, which was made available to the researcher in cooperation with the mobile telecommunication company (Hamrahe Avval).

These data contained statistical information on network quality measurement indicators at the covered points. Since the cellular network was very dynamic and complex and the volume of generated data was very high, we started the work by choosing one of the worst base station controllers in Tehran. This controller is located in District 8 of Tehran and covers a considerable part of this area. The data contained 64,114 records. Among the 70 indicators and features of this data, 14 indicators were selected by experts to achieve the goal of this research. The characteristics and impact of each on the network's quality and the cause of the occurred fault in the network were investigated.

Table 2 provides the evaluated the statistical characteristics of these indicators. Since the values of different indices have different distributions, the minimum–maximum normalization method was used to normalize the data.

**Table 2.** Statistical status of the used indicators.

| The Indicator's Name | Mean | Standard Deviation |
|---|---|---|
| CSSR | 0.950793 | 0.106 |
| CDR | 0.0412 | 0.068 |
| HSR | 0.832 | 0.2735 |
| HTr-Rate * | 0.3408 | 0.4048 |
| SDCCH-Congestion-Rate | 0.0555 | 0.18109 |
| TCH-Congestion-Rate | 0.0401 | 0.11137 |
| SDCCH_DR | 0.0448 | 0.1404 |
| SDCCH-MHT | 0.0707 | 0.076 |
| SDCCH-Assign-Success-Rate | 0.4676 | 0.4249 |
| TCH-ASSIGN-FAIL-RATE | 0.0478 | 0.0672 |
| RX-QUAL-DL | 0.8485 | 0.1394 |
| RX-QUAL-UL | 0.4080 | 0.4847 |
| Random-Access-Success-Rate | 0.6765 | 0.4295 |
| TCH-Availability | 0.9366 | 0.1233 |

* The HTr_Rate feature is the ratio of the traffic carried on the half-rate channel to the total traffic of that station.

The histogram of the CDR call drop rate indicator, which shows the drop rate in calls, is shown in Figure 7. As is clear in the figure, for some samples, the value of this index was higher than the threshold and, therefore, it indicated a problem in the records.

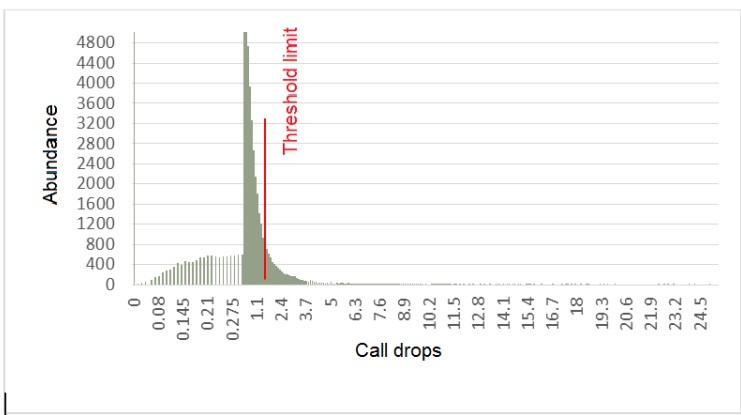

**Figure 7.** The histogram of the dropped calls in the data.

*3.2. The Definition of Quality Criteria and Indicators Analysis*

After selecting the features and pre-processing the data, the criterion for checking the quality status of the network was introduced. The evaluation criterion was the CCC index, which is obtained from the multiplication of a number of the most important indices. The histogram of this index was derived from 64,115 records and is illustrated in Figure 8.

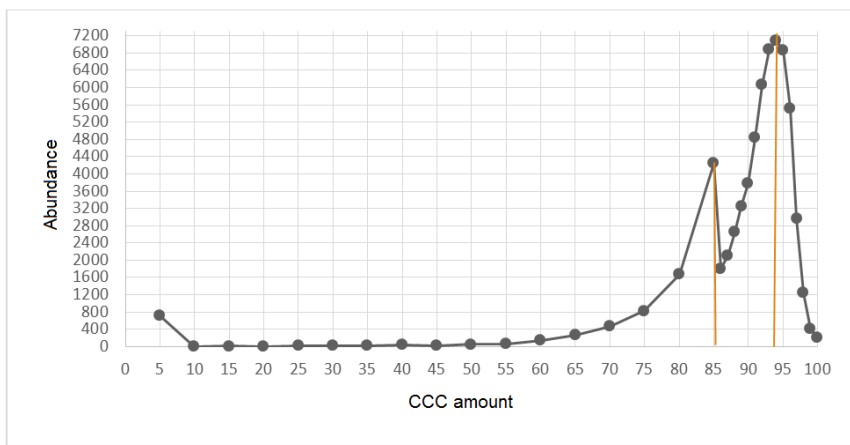

**Figure 8.** The histogram of CCC index value in one month of the examined data.

As expected, and as can be seen in the above diagram, the range of numerical values of CCC was between 0 and 100. Values close to 100 indicated higher quality. In Figure 8, there are two knees. Therefore, we divided the network status into three discrete categories. The first category had a CCC value higher than 94 and represented excellent quality. The second category had a CCC between 94 and 85 and represented acceptable quality. Several records fell into the third category, having a CCC less than 85. This category represented unacceptable quality of the network, and, therefore, our work priority was to focus on the data of the third category, the unacceptable category. From now on, we considered the records with CCC less than 85 as faults and identified the cause of the fault in these records.

Before addressing the issue of identifying the cause of the fault, an analysis of the values of these indicators was conducted. To do this, we used classification algorithms and, especially, a decision tree. The reason for using the decision tree algorithm is this method's display ability and its good interpretation ability. In decision tree algorithms, the Quest method is used. This method had high accuracy for our provided data and this accuracy was obtained at a lower tree depth. Therefore, it avoided the problem of overfitting the model on the training data. The following tree separated the test data with 93.66% accuracy. The results were obtained with the help of Clementine software. It can be seen that among the 14 features used by experts in these networks to analyze the quality

status, and among the seven indicators used in defining the quality criteria of the network, only three indicators, those of call interruption, interruption in the signaling channel, and the success rate in achieving the signaling channel, could estimate the network status (Figure 9).

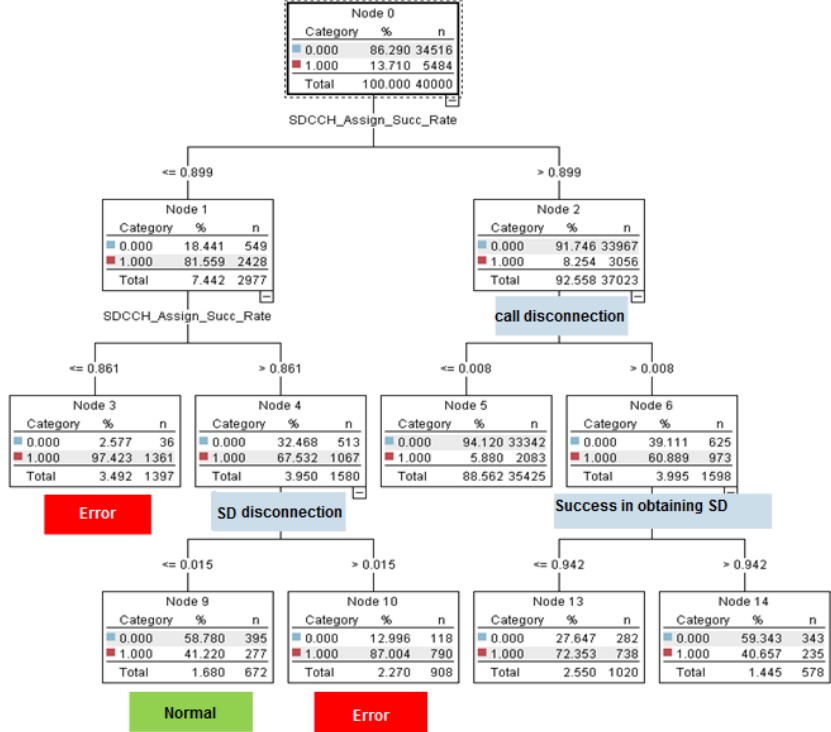

**Figure 9.** A schematic of the data production tree with two fault and non-fault categories by the use of Quest algorithm.

### 3.3. Clustering Analysis and Fault Cause Identification

In this section, we examine the results obtained from the clustering performed on the performance support system data. In clustering, the data is generally divided into traffic and signaling data categories. The results of each type and their analysis are given in the following. First, the traffic clusters were checked, and then the signaling clusters were studied. As mentioned before, the characteristics used in the traffic section included:

- The percentage of communication failure in the TCH channel.
- The percentage of congestion in the TCH channel.
- The percentage of loss in adopting the TCH channel.
- The average quality of the uplink and downlink signals.

The clustering results of different methods are shown in Figure 10.

As illustrated in the above figure, the Expectation-Maximization clustering method provided a better response to the data than other methods. In addition, the optimal number of clusters was 5. Therefore, the EM method was applied to the data with several 5 clusters. The information related to this clustering was provided to the expert, and the expert comments, confirming the correctness of the obtained clusters, are given below, along with a view of the data distribution in each cluster. The data detected as faults were divided into traffic and signaling categories, and, as mentioned, the optimal number of clusters for the traffic data under consideration was 5. Therefore, one of these clusters indicated a favorable situation in the traffic cluster, which meant that the problem was only in the signaling data, and the system had no problem in terms of traffic.

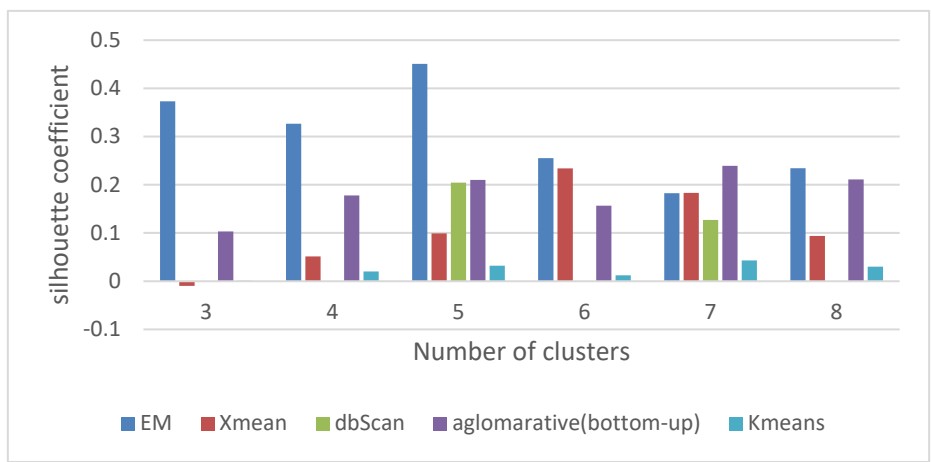

**Figure 10.** Silhouette coefficient according to the number of clusters for different clustering methods–traffic section.

A similar situation existed for signaling data. The problem was that some records only had traffic issues (Figure 11). The allocated records in (c) suffered from high outliers in the SD channel and the capacity problem. SD channel congestion and outages are of great importance, due to their direct impact on the shared network experience. Some influential factors in increasing the absolute value of this channel are the coverage level, interference, low quality, and instability of communication links in the Abis interface. Since there was a congestion problem in this cluster, the probability of a coverage problem in this area was less than the probability of other reasons. On the other hand, the cut-off value in this cluster was much higher than the threshold. Therefore, the cause of the problem might be hardware problems. The congestion problem was very critical in (d). The behavior of the channel access success indicator was also unacceptable. Therefore, this cluster's record had the problem of inadequate coverage, as well as low signal strength. Furthermore, these sites should be checked for interference. Cluster (e) had the same problems as (d), in terms of high cut-off value and failure to access the channel, but the difference was that there was no congestion problem in this cluster. Finally, cluster (f) had a hardware problem, due to the wildly inappropriate success rate of reaching the signaling channel.

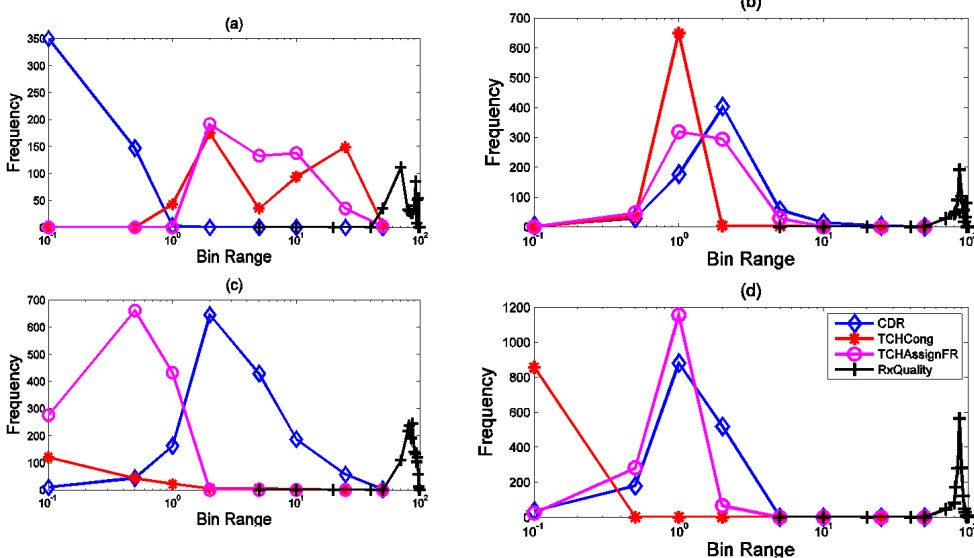

**Figure 11.** Indicators of statistical distribution of the signaling sector. Each cluster represents a mode of the network.

### 3.4. Evaluation of the Results

The expert was asked to label the new data records according to the obtained clusters to evaluate the results from the records clustering. For this purpose, traffic and signaling data documents were provided to the expert separately. Having the labels for the obtained clusters in the previous step, it was now possible to test the accuracy of the clustering results by running different classification algorithms and using the test data labeled by the expert.

Therefore, different classification algorithms were applied to the data in both traffic and signaling data parts. The evaluation results of the other models on the test data are given in Table 3. It should be noted that, due to the lack of balance in the training data in the records related to each class, Clementine software was used to balance the records of each type. This software was also used to apply different classification methods to the data. The number of training set records was 8006, and the number of traffic and signaling data test set records was 53.

**Table 3.** Evaluation of obtained clusters for traffic data.

| Band Category Type | The Accuracy on Data Test |
| --- | --- |
| Neural Network–Rapid | 86,54 |
| Neural Network–RBFN | 78,85 |
| Neural Network–Dynamic | 80,77 |
| Quest decision tree | 59,62 |
| Support vector machine | 82,69 |
| Regression- band category tree | 57,69 |

By obtaining the different classification methods' results, accuracies of 90.38%, 86.54%, and 78.85% for traffic data were obtained from some forms of summarizing opinions, such as voting, weighted voting concerning model confidence, and the highest confidence method. The standard voting method worked better on our data. The results of the used methods for the data of the signaling section are shown in the following table. The voting method is the best way to collect opinions, with an accuracy of 92.31. Other practices, including weighted voting, had an accuracy of 88.46 and 80.77 in relation to model confidence and the highest confidence (Table 4).

**Table 4.** Evaluation of obtained clusters for signaling data.

| Band Category Type | The Accuracy on Data Test |
| --- | --- |
| Neural Network–Rapid | 80,77 |
| Neural Network–RBFN | 88.46 |
| Neural Network–Dynamic | 78,85 |
| Neural Network–General prune | 80,77 |
| Quest decision tree | 59,62 |
| Support vector machine | 90,38 |
| Decision tree categorizer–regression | 71,15 |

### 3.5. A Comprehensive List of Identified Fault Causes of the Performance Support System Data

In the following part, a summary of the causes of faults in the cellular network is provided. The results obtained from numerous conversations with industry experts in the mobile network optimization department of the Mobile Communications Company (Hamrahe Avval) were collected.

**Reason number 1:** hardware problems, including transmitter failure, antenna feeder failure, and combiner failure: the impact of these types of fault can be seen in call and indicator interruption in the signaling channel, failure rate in traffic channel assignment, and in the signal quality.

**Reason number 2:** hardware problems related to the link, containing several modes. The first case is when the connection is completely disconnected, and the site is turned off. The second mode refers to the time when the link is momentarily down and, therefore, has a negative impact on the traffic channel availability index. The third case is related to a type of problem in the system that cannot be traced by the value of the indicators and can be identified according to the drive test (which is examined in detail in the next section). In this case, the channel availability indicator does not indicate a problem, but the signal quality suffers due to the high Bit fault rate (BER). In some cases, this problem may also be reported by subscribers.

**Reason number 3**: Hardware problems caused by Abis interface failure. This happens when the number of signaling channels is insufficient compared to the region's demand and subscribers' demand. Therefore, the impact of this problem can be seen in the SDCCH-Assign index.

**Reason number 4**: problems caused by overshooting. Overshooting is caused by improper design of sites and inappropriate coverage of a place in remote areas, and can be found by checking the value of the advanced timer. To solve this problem, it is necessary to change the physical characteristics of the antenna, such as its angle, height, and direction.

**Reason number 5:** frequency interference, including interference in the BCCH channel and traffic channel. If the interference is in the BCCH channel, the traffic parameters are also affected in addition to the signaling parameters. If the interference is in the traffic channel, only the traffic parameters are damaged.

**Reason number 6**: lack of capacity problems in the traffic and signaling channel. The solution is to increase the capacity and define relevant features in the settings of cellular networks to allocate dynamic capacity to the required channels according to the network traffic situation.

In this section, the method used to detect the fault and identify the cause of the fault in the network is examined in detail. The difference between this work and other similar works in this field can be divided into the following. First, unlike the vast majority of similar works, our method used actual data to identify the cause of the fault in the network. Other similar works cannot model the mode of the network. Our proposed model considers all the essential statistical characteristics and indicators of the network at the same time. This means that, unlike all the mentioned methods in the literature review section, it has a comprehensive view of the network and the fault types that occur in the network. Finally, the proposed model has the most negligible dependence on expert opinion. This is even though most of the proposed methods depend on the expert to build the model and adjust the parameters. The expert's opinion in setting the parameters of the model may have a significant impact on the results obtained. Due to its negligible dependence on expert opinion, our model has the advantage of avoiding such significant impacts.

*3.6. Drive Test*

As mentioned in the first section, the second category of the used data in this research is the data related to drive test measurements. The drive test is a real test of the network to obtain detailed information about the location of the fault, how the antennas around the location signal, and to check the redistribution status in the routes. To perform this test, the operators send a group of human resources to the site. The responsibility of human resources is to check the network quality from different perspectives. Various scenarios, such as short calls, long calls, and non-calling mode or idle mode, are proposed by human power. The non-calling mode analyzes the mobile device's signal exchanges with the receiving station. The test is a method to measure and check the coverage status, network capability, and quality from a common point of view. The test is conducted by making

a call through several mobile phones with software to control and record the reported measurements and parameters from the BTS side and the Global Positioning System (GPS). The measurements are transferred from the mobile phone to the computer. The measurements only report network mode from different perspectives and cannot identify the fault type and cause. In general, the drive test process consists of the following steps:

1. Setting the devices for the required measurements
2. Defining routes
3. Determining the testing time and type
4. Performing the test
5. Collecting data and related reports

Figure 12 shows the scope of conducting a drive test according to the division of urban areas and the boundaries between the central controller stations.

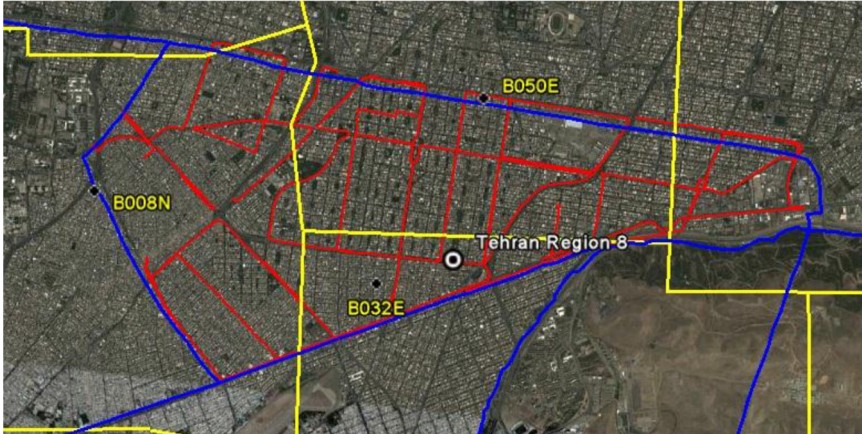

**Figure 12.** Drive test route map according to the division of urban areas and the border between base controller stations. The blue lines illustrate the 8 urban areas, the yellow lines illustrate the border between the base control stations, and the red lines illustrate the drive test route.

By analyzing the conducted measurements in the drive test, some results, such as determining blind spots and non-coverage, determining the displacement of the sector or the feeder, determining an area with interference, interruption of conversation, and failure in the transfer, are determined. In general, the identified faults list in the drive test is as follows:

1. Problems related to sector and feeder changing
2. Call interruption problems caused by interference and sector signal overshooting.
3. Interference problems that require checking assigned frequencies to nearby cells.
4. Problems related to outsourcing, which can be traced by adequately setting the neighborhoods, coverage area, and related parameters to outsourcing.
5. Problems related to the coverage area, which, by adjusting the height, power, and angle of the antenna, lead to a change in the design and, finally, the installation of a new site.

*3.7. Definition of Different Scenarios in the Drive Test*

In general, three different scenarios test the network condition in a drive test. Each of these three scenarios examines the network status from particular aspects, and various faults are identified in these scenarios. In the following, we describe each of the three scenarios.

**Scenario number 1:** Idle mode: In this situation, the responsible person for the test puts the mobile phone in idle mode. This means that phone calls cannot be made. This is to test the network in serving cell reselection when moving the mobile phone. In addition, the

received signal level is measured at different points of the network, and the locations with suitable signal levels are provided to the optimizer for further investigations.

**Scenario number 2:** Short calls: The short call scenario means evaluating the network in terms of call establishment parameters. That means assessing whether the mobile phone subscriber can easily make a call, and what the problems in the process of making a call are, including access to the signaling and traffic channels. Call blocking parameters and thriving network access rate are investigated in this scenario.

**Scenario number 3:** Long calls: The long call, unlike the short call, may be maintained continuously during the drive test. This scenario aims to test the network in terms of call interruption rate, conversation quality, received signal quality, and check reassignments.

This article examines the problems affecting network service quality by reviewing all three scenarios. In the following, we first introduce the measured parameters in the drive test and then describe the process of identifying network problems in these three scenarios.

*3.8. Introducing the Measured Parameters*

This section introduces the essential measured parameters in the drive test that form our feature vector. This information is used to design the network; for example, to achieve the correct redistribution and to control the radio signal's power.

**1. Signal quality:**

In some systems, such as UMTS, the signal quality parameter is directly dependent on the signal-to-noise ratio [45]. In other methods, such as GSM/GPRS, the signal quality is a Bit or symbol fault rate function.

**2. Signal level:**

The received signal level is a measure to control the power of the received signal and the transmission at the place of receiving the signal. The accepted signal range by the mobile user is between $-110$ dbm and $-48$ dbm. The low level of the received signal can have several reasons. For example, consider a rural area where the distance to the base transmitter and receiver station is not suitable. This area receives a weak signal, due to the long distance to the BTS, and installing a new station is impossible due to high cost. Another reason for the decrease in the signal level can be the presence of harmful interference with other received signals in the area. In addition, some tall buildings cause a rapid attenuation of the signal strength and, therefore, the signal inside the building is not strong enough. An unacceptable signal level can lead to call termination [46].

**3. Speech quality:**

The literature on cellular networks measures speech quality with the SQI-MOS parameter. In its calculation, an algorithm is used to consider the network mode, such as frame fault rate (FER), and bit fault rate. The algorithm can also calculate the coding type used in speech transmission, which affects the speech quality level and the highest possible quality, by calculating the difference between the analyzed sound quality on the side of the transmitter and the receiver. The result is shown as a number between 1 and 5.

**4. Carrier-to-Interference (C/I) Ratio:**

One of the most effective and accurate methods for analyzing interference in communication channels is measuring the Carrier-to-Interference (C/I) Ratio. Route design and equipment design are two critical factors affecting the interference level. The neighborhood of nearby systems that use the same frequency band is one of the apparent reasons for interference, due to the wrong design of routes.

**5. Advanced Timing:**

In the literature in this field, advanced timing refers to the time it takes for the mobile phone signal to reach the base station. Since GSM uses multiple access technology, based on time division, to share a frequency band between different users and considering that the users are located at an extra distance from the base station, this criterion can be used to estimate the distance of the user to the base station used.

### 3.9. The Architecture of Working with Drive Test Data

As mentioned in the previous section, to review the architecture of working with performance support system data and use the received data through field testing, it was necessary to take measures to identify the meaningful patterns from the data, according to the illustrated architecture in the following figure. Therefore, after collecting the raw data of the drive test, through the TEMS Investigation software, which was used to analyze these data, a report of the network mode was prepared, based on the scenario type and the characteristics and effective indicators by the use of expert opinion. These features were fully introduced in the previous section. To use the extracted data from this software, it was necessary to convert the data format into a usable format for the next steps. At this stage, we used MapInfo software to transform data into a.csv format for data mining and meaning recognition algorithms. The obtained data needed to be pre-processed. Therefore, it was necessary to clear this data before performing any operations. Then, appropriate algorithms were applied to the data to identify the meaningful patterns in the data, and, finally, the results were validated according to expert opinion. If there was a need to make changes in the model, the changes were applied, and again we used the expert to evaluate the results (Figure 13).

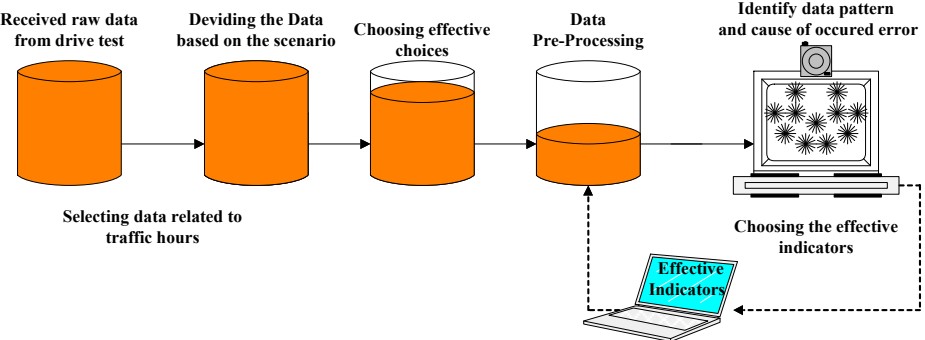

**Figure 13.** Architecture of working with drive test data.

### 3.10. Data Preprocessing

As mentioned in the previous section, there was a need for data pre-processing to obtain results with higher accuracy. Missing values and records containing outliers were removed from the data. Outliers were removed from the data by identifying the allowed values for each attribute. The used data in this section were ordinal data types. Ordinal discrete numbers are the same as categorical data, with the difference that the states in ordinal variable values have a meaningful order. To manage ordinal data, it is necessary first to establish correspondence between each data value and its corresponding rank so that the data falls in the interval of [1, Mf], where Mf is the number of different modes in the ordinal data for feature f. Since there are many other states for ordinal data, it is necessary to normalize the data to the interval [0,1] to prevent all features from having the same effect. So, if the i-th data rank in attribute f is rif, we have:

Equation (7): Normalization of ordinal data

$$z_{if} = \frac{r_{if} - 1}{M_f - 1} \tag{7}$$

To calculate the degree of similarity between two data records, it is possible to do the same for continuous data [10].

It should be noted that the available data that indicate the overall situation in a network are usually unbalanced because the network works well in most areas. There is a smaller percentage of points that have problems. Since the optimal range of all the used features in this research was included in the standard, we could separate the records in which the optimal degree of all variables were observed from the other data and label them as fault-

free to investigate the problem of data imbalance. Therefore, by doing this, a significant amount of data was reduced, and would be more suitable for analyzing the cause of a fault in the network.

### 3.11. Identifying the Data Pattern and Identifying the Fault Cause

To divide the data into meaningful categories, it was necessary to apply different clustering algorithms to the data, according to the examined scenarios and the results obtained from the evaluation of the clusters. The best separating algorithm should be identified from the drive test data. Since the data of this part was similar to the data of the last section, it was raw data, and it was impossible to use external criteria to evaluate the correctness of the obtained results. It was necessary to use internal performance evaluation criteria to assess the obtained clusters. The standard for evaluating the clustering of the drive test data was also similar to the performance support system data, namely, the Silhouette coefficient. It should be noted that to ensure the ability to cluster the data, the Hopkins coefficient was applied to the data of each scenario. The obtained result for the scenario of the idle mode was 0.88%, and for the long call it was 0.94%. Therefore, the data had good capability for clustering. The analysis of the obtained results, according to different scenarios, is given in the following part. Table 5 summarizes the related information to the features examined in this test.

**Table 5.** A summary of the statistical information of the used features.

| Feature's Name | Minimum | Maximum | Optimal Interval (Standard) |
| --- | --- | --- | --- |
| Carrier-to-interference ratio | 5 | 25 | $\geq 25$, <35 |
| Signal level | −101 | −31 | More than −65 |
| Signal quality | 0 | 7 | Less than 3 |
| Call quality | 0 | 4.1 | Equal to 3.9 |
| Advanced timing | 0 | 3 | Less than 1 |

Scenario number 1, Idle mode: This tests the received signal level at different points of the network. Locations with inadequate signal levels are reported to the optimizer for further investigation. According to the characteristics that were considered in this scenario, including the signal level, the ratio of the carrier power to the interference and the advanced scheduler, and the number of optimal clusters, which were of three numbers, according to the expert's opinion, after data pre-processing, we had the results indicated in Figure 14.

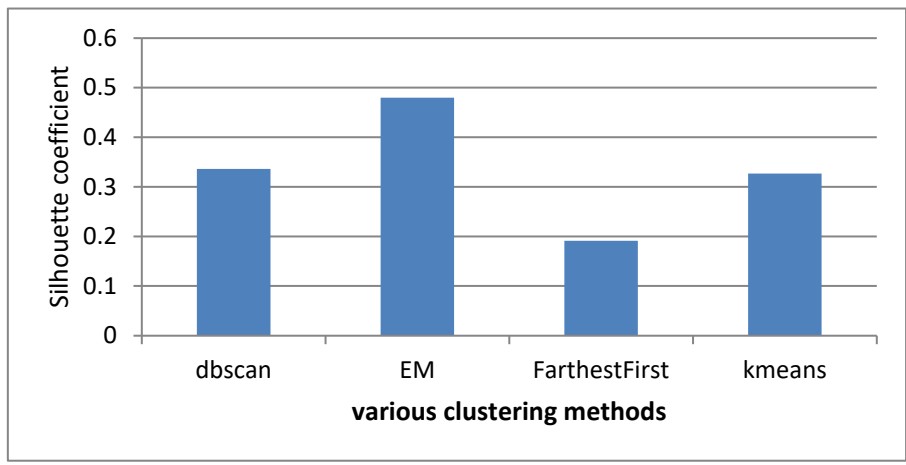

**Figure 14.** Silhouette coefficient on the idle mode scenario with the number of 3 clusters.

Therefore, on the drive test data in the idle state scenario, the estimation maximization (EM) method provided a more suitable answer than other methods. A more complete description of the obtained clusters' characteristics is given in Table 6.

**Table 6.** The details of Clusters and Feature's name.

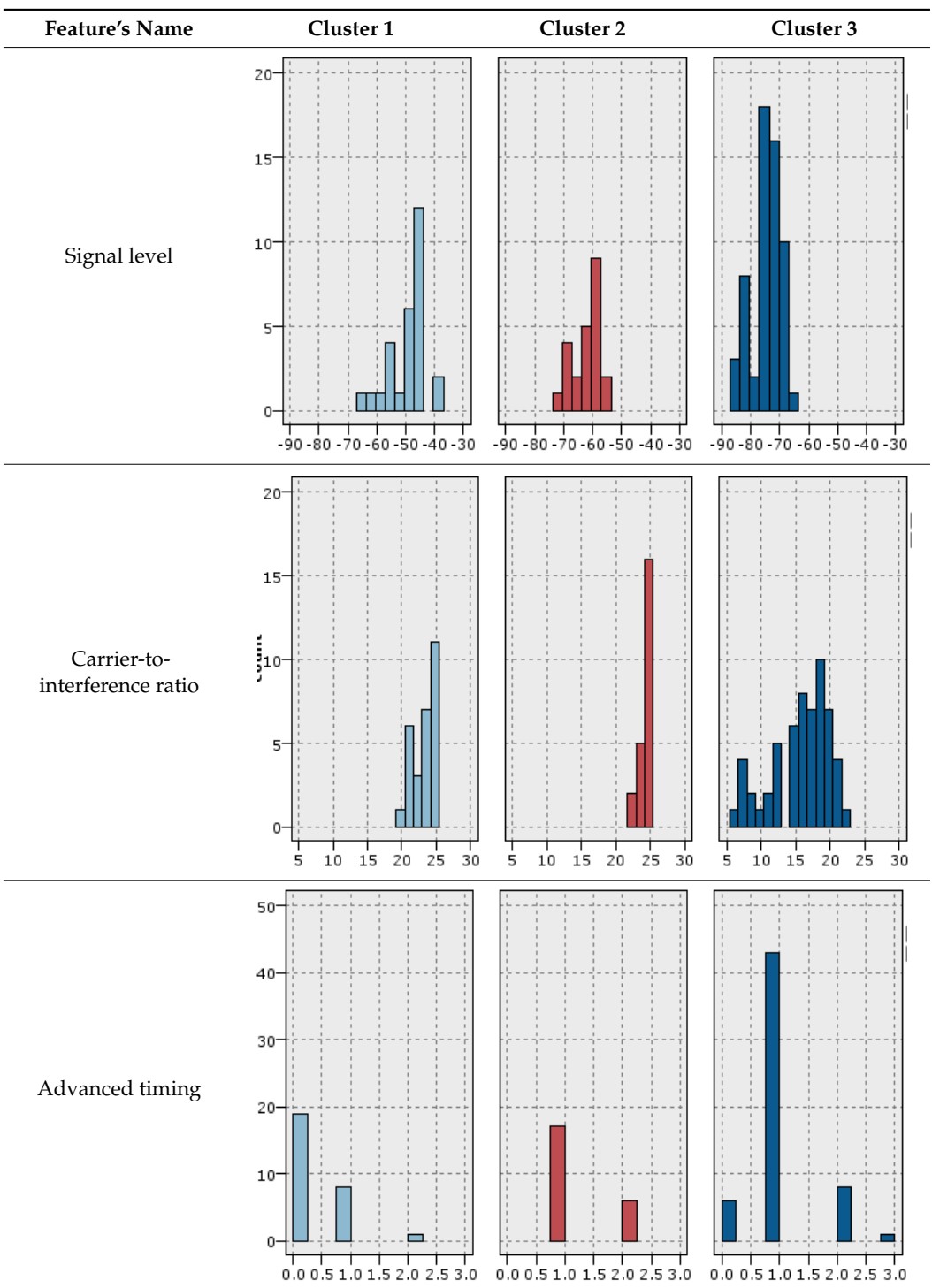

Each of the obtained clusters indicated good, average and unfavorable network conditions in terms of the received signal characteristics of the mobile phone in the idle mode. In cluster number 1, according to the statistical information table of the used features, all records were in a suitable and optimal condition. In cluster number 2, although the inter-

ference in the environment was negligible, the signal level was weaker than the optimal mode. The low signal level in the records included in this category was due to the relatively large distance between the base transmitter and receiver station. Since there was very little interference in this category, these areas did not have the problem of improper frequency design of adjacent cells or the problem of interfering signals from neighbors. Therefore, the records of this cluster did not report a specific problem in the network. If a new antenna was installed to improve the signal level in this area, it would interfere with the received signal in the neighboring points. The third cluster contained records that were not far from the service station and had an unacceptable level of interference and signal. This meant that the path obstacles might reduce the signal strength. The interference caused by the signals of the neighboring cells should be corrected by resetting the site parameters, such as the height and angle of the antenna in the adjacent cell.

Scenario number 2, short call: The purpose of this scenario was to check the network in terms of its access success rate. Many calls were attempted in this scenario. In the available data of this research, there were 8 blocked calls out of 349 calls in the short call scenario. These results showed high interference in these areas, and the signal level was insignificant. Further investigation of the reasons for blocked calls was available in performance support system indicators and is covered in Section 4.

Scenario number 3: Long-term call: This scenario, which was the most critical scenario in the field test, measured signal quality, conversation quality, carrier power to interference power ratio, and advanced timing. The purpose of this scenario was to identify the state of the system on call interruptions and handovers.

As seen in the following figure, among the different clustering methods applied to this data, the k-means method achieved a higher Silhouette coefficient than other methods. Therefore, we first briefly overview the k-means clustering method. The details of the results are given later in this section.

The k-means method is among the best cluster partitioning methods. This means that, according to the number of clusters, there is an attempt to optimize the data division according to the similarity or distance criterion (Figure 15). Several algorithms with a different number of clusters were tried on this data. For the k-means method, the number of 4 clusters had the best answer. The Silhouette coefficient value for this cluster was 0.6269.

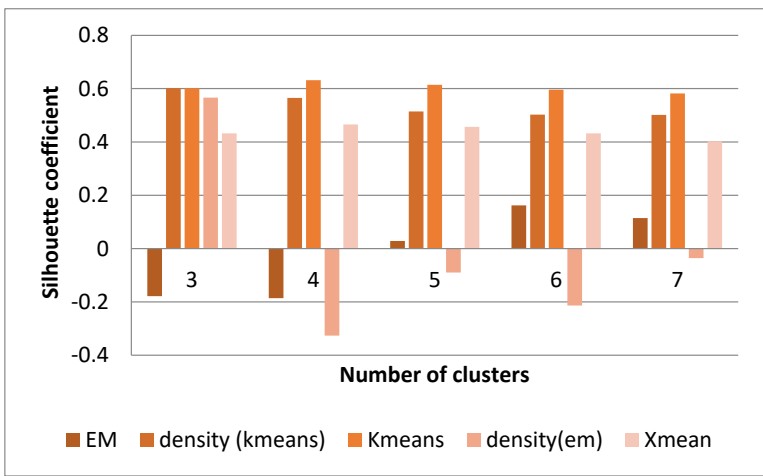

**Figure 15.** Silhouette coefficient value of different clustering methods–long call scenario.

Table 7 illustrates the obtained histogram of the features in 4 categories.

The signal quality in cluster number 1 was inadequate. Low signal quality was due to low power and signal level. Paying attention to the characteristic value signal power–interference power ratio showed significant interference in the system due to the signal level. On the other hand, since the value of the advanced timer parameter was low, the decreased signal level was not the long distance from the transmitter station. Therefore, this

problem could be seen in the voltage standing wave ratio (VSWR). The problem, expressed as a loss in signal strength, was due to hardware problems in the antenna or its incorrect installation. Another reason for the problem was the wrong tilt of the antenna angle. It could be seen that in this situation the quality of the signal was not good, and the received sound in the receiver had good quality. This problem could be seen in the audio coding type used in this network.

**Table 7.** The histogram of different features in the obtained clusters–long call.

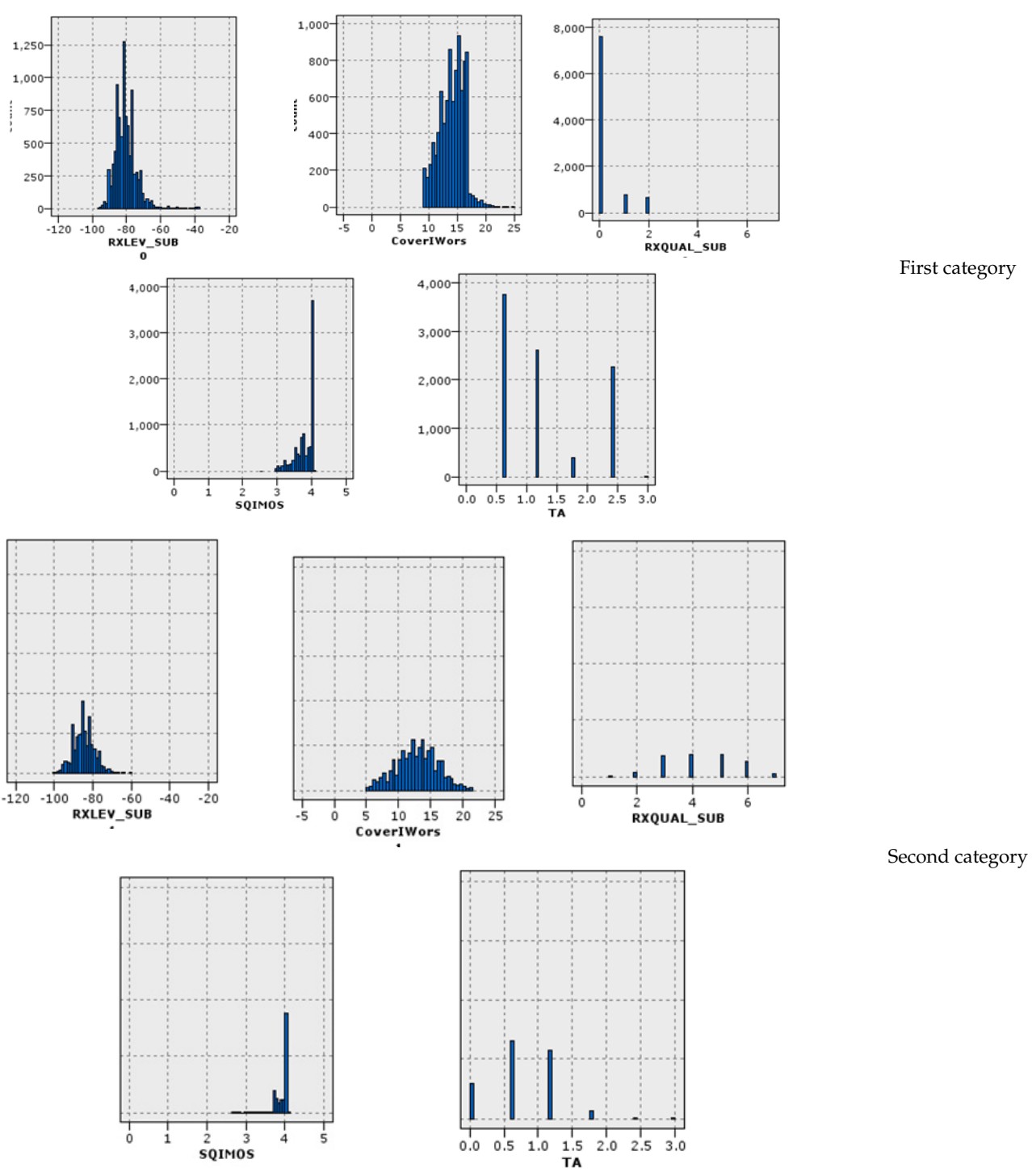

First category

Second category

**Table 7.** *Cont.*

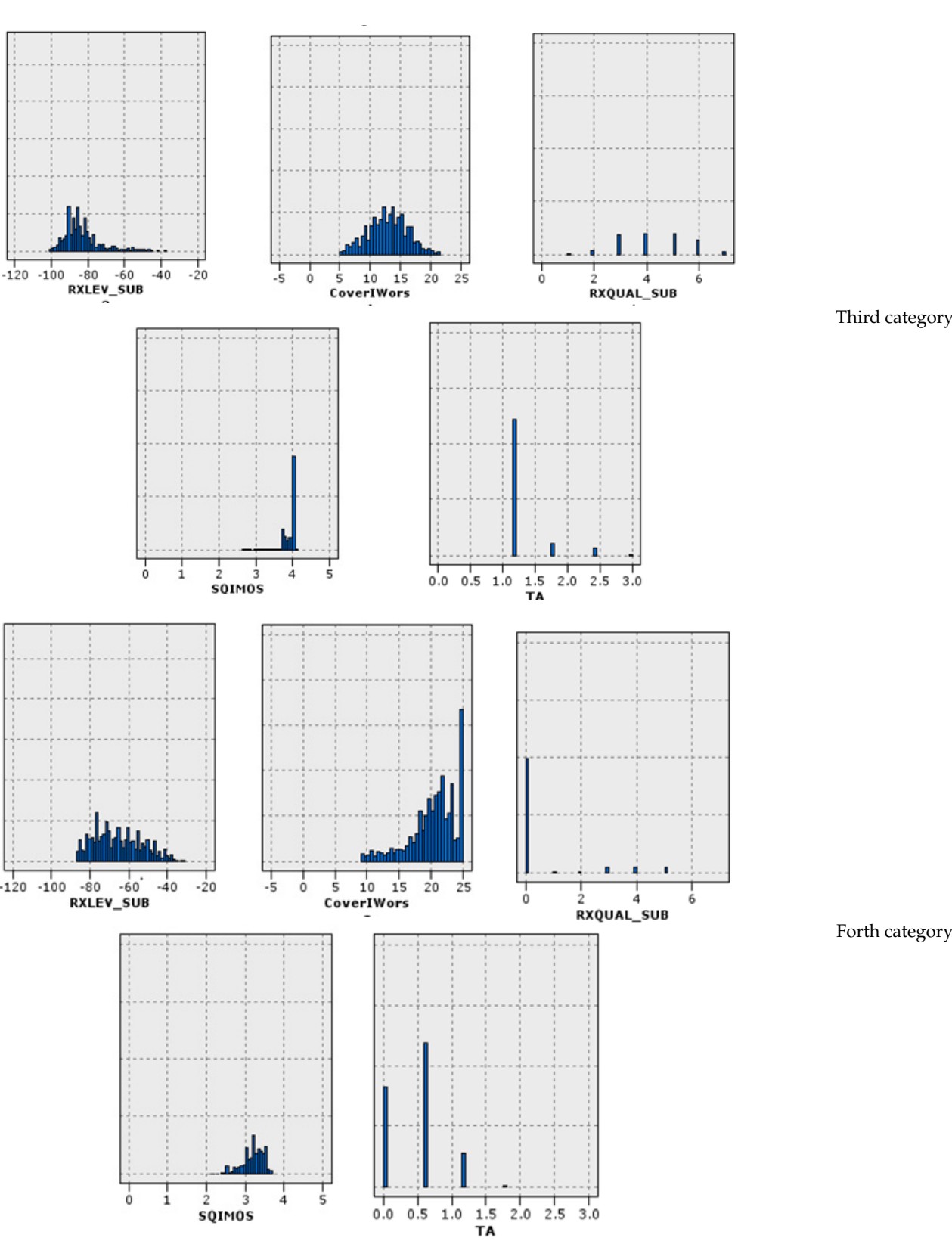

Third category

Forth category

In cluster number 2, the signal level was low. A high value of the advanced timer confirmed the presence of a problem in the standard received signal. This meant that this signal reached the subscriber from a long distance, and, therefore, the service site's signal

was out of range. Thus, site design parameters, such as antenna height and angle, should be considered to reduce the coverage area or the antenna power.

In cluster number 3, all features were in good condition. This cluster showed the proper state of the network.

In cluster number 4, the bad quality of the signal was not caused by the power reduction and signal level or interference. On the other hand, the value of the advanced timer also showed the system's status. Therefore, according to the expert, the reason for this was hardware problems in one of the sources of sending and receiving information in antennas called TRX.

Cluster number 5 showed high interference in the system. On the other hand, the signal level in this cluster was suitable. Therefore, the interference strength could be seen in the low power of the carrier signal. In this cluster, the advanced scheduler also had an acceptable value. Therefore, the reason for the existence of this problem could be attributed to the presence of interference from external sources, such as signals from other operators, or incorrect installation of sectors and the need to swap them.

### 3.12. Review the Handover

As mentioned before, one of the issues addressed in the long call scenario is the handover issue. Our model investigated this issue separately. By reducing the quality of the signal received by the user, which is a function of the bit fault rate, the network is obliged to transfer the user's call to a better cell. Sometimes, outsourcing may happen later than the appropriate time, and this phenomenon leads to common dissatisfaction with the network.

To solve this problem, the proposed solution is to use a time window to detect the quality and level of the signal received by the user in a long call. The time window length and the threshold value were considered on the quality and level of the signal according to the expert's opinion. If the subscriber received signal quality worse than, or equal to, 5 and the signal level was less than $-75$ within six times of receiving information from the central transmitter station, our system requested retransmission in the network because otherwise the user's call would be disconnected, which would lead to user dissatisfaction. Figure 16 shows the signal quality and level at different network points.

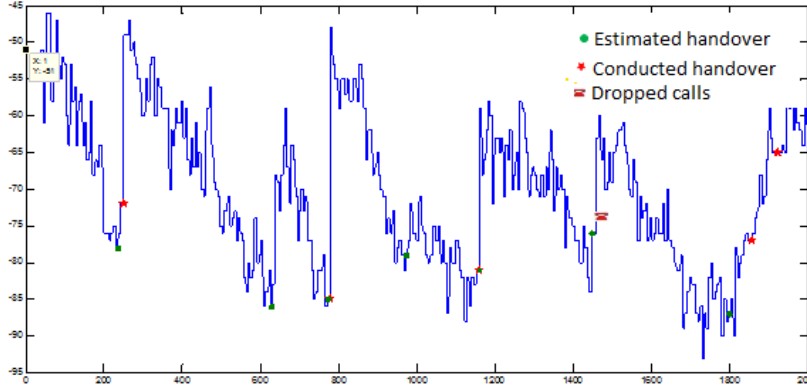

**Figure 16.** Silhouette coefficient value of different clustering methods–long call scenario.

This section introduces the drive test, one of the scientific and engineering methods for identifying the location of the fault and the cause of its occurrence. Different scenarios performed in a drive test by a team of experts were investigated separately in this research. For the scenario of idle mode, three separate signal strength categories were considered for the subscriber, which divided the network mode into three categories: good, average, and bad. In the short call scenario, where the network mode was examined in terms of the call establishment parameters, after identifying the exact location of the blocked calls and the state of the received signal, the characteristics of the indicators of the performance monitoring system were used to analyze the cause of the problem. Finally, the long call

contact of four categories was obtained for the most critical scenario. Their interpretation is available in detail in Section 3.7 To evaluate the clustering method, in addition to the Silhouette coefficient, data classification using the cluster number label was used, the accuracy results of which are given in the training and test data at the end of Section 3.7 Relocation, one of the most important goals of long call scenarios, was also examined separately in this section. The results of the handover evaluation are given.

### 3.13. Combining Data Sources

The purpose of this section is to combine the results of the second and third sections. In other words, in this section, we wanted to complete the architecture of automatic fault detection and detection in cellular networks by examining the results of these two sections. As mentioned in previous sections, two critical data sources for fault detection in cellular networks are performance support system data and field test data. Each of these two data sources examines the network status from different aspects. Therefore, the qualitative issues of the network were discussed from two different perspectives related to these two data sources. To identify faults and problems in the network, it was necessary to examine these data sources separately.

Another data source that we used in this section to more precisely identify the fault cause was data related to subscriber complaints. These data, which were collected from the communication center with the subscribers of the mobile telecommunication company (Hamrahe Avval), indicated the problems reported to the operator by the subscribers of this network. In the following, we briefly explain customer complaints' data and then introduce the data combination methods. The results of combining these three data sources are given at the end.

### 3.14. Subscriber Complaints Dataset

Investigating customer complaints is one of the essential activities of customer-oriented organizations. Considering the competitive world among operators, it is necessary to manage customer complaints and adequately satisfy customers. The block diagram of the communication of subscribers' complaints with different departments of the mobile telecommunication company (first companion) is shown in Figure 17.

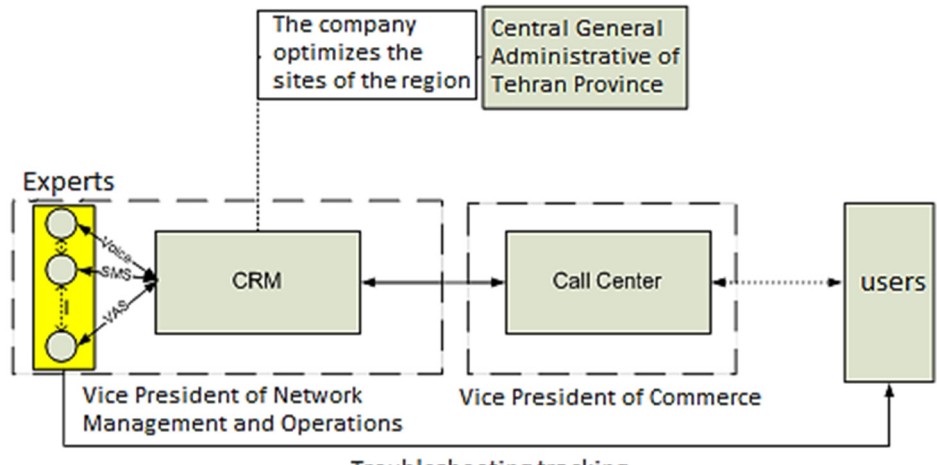

**Figure 17.** Silhouette coefficient on idle mode scenario with 3 clusters.

After the customer communicates with the call center, the user report is separated from other calls based on the communication with the technical department. At this stage, complaints related to a similar incident are consolidated before being sent to the technical department and, finally, sent to the customer relations department of the technical department. The received reports in the technical department are re-checked and divided into categories, such as Voice, SMS, etc. Problems related to the optimization of regional

sites and the maintenance of BTS sites are sent to the Central General Administration of Tehran Province. After investigating the problem, the expert's report is transferred to the call center and the call center contacts the subscriber if necessary. The available data collection we used of to complaints from mobile telecommunication center subscribers related to technical problems, and was reported in October and November of 2020. The number of records of this data was 3200 in total. Table 8 below shows the features in this data.

**Table 8.** The data characteristics of subscribers' complaints.

| Feature Name | Feature Type |
| --- | --- |
| Registration time | Time |
| Request type | Text |
| Request description | Text |
| Site ID | Text |
| BCS ID | Text |
| Explanation | Text |

Among the reported problems by subscribers, temporary network disruption, echoing, receiving false messages during calls, coverage problems, disconnection, and antenna weakness were mentioned. The total number of these problems was divided into 6 general categories; about 80% of the reported issues was related to antenna and network coverage problems.

This category's data was used to help better identify network faults, especially those not specified in the other two data categories. Therefore, by adding this information to the previous data, this information could be used to increase the accuracy of the proposed model. Our observations showed that about 1% of all calls raised as technical complaints to the mobile telecommunication company were unrelated to the technical department. The problem was related to the subscriber himself or herself, for reasons such as problem registration by non-technical personnel in the operator, as well as issues associated with the user, such as problems related to some types of mobile phones (for example, some mobile phones cannot support the AMR feature), wrong settings on the shared telephone (for example, call transferring to the wrong number), the existence of a hardware problem in the phone that led to poor quality audio, etc.

Among the remaining 99% of complaints, about 35% of these calls stated problems were not worth taking any appropriate action, for technical and economic reasons. Calls were mostly related to the lack of antenna in remote areas or certain parts of a private house; for example, a user in a village with a relatively small population may contact the customer complaint center for lack of coverage in the network. Since there would be economic problems in installing a new site in that area, the mobile telecommunications company cannot take action to satisfy the customer. According to discussions with the experts, no specific action is taken to improve the network situation for this category of reports. Finally, 65% of the subscribers' complaints reflected a real problem in the network.

It is important to note that all customer complaints should be investigated. After investigating a common complaint, according to the amount of other data available, such as performance support system data and field test data, appropriate actions are taken for that data if necessary. In other words, if both other data sources are available, if the other two data sources confirm the absence of a problem in the area, that problem is not transferred to the next step. In the case of a severe problem in the network, one of these two data sources could transmit this problem.

Our reviews of the data confirmed the validity of customer complaint data. For example, on the 15th and 16th of October, several subscribers' complaints were recorded in the system for a specific cell in the network. The received information from the performance

support system for this particular cell confirmed the occurrence of a problem in the area, because the average CCC of this cell was below 85 during high traffic hours, and our model detected the problem. Therefore, paying attention to subscribers' complaints is one of the critical issues in identifying causes offaults.

*3.15. Fault Detection*

According to the conducted studies and the results presented in the second and third sections, which specifically focus on the performance support system and drive test data, we offer a mechanism to combine this information to complete the architecture of automatic detection of recorded fault.

All three of our data sets were unavailable at all times, and, due to drive testing limitations and subscriber complaints, the methods used to combine information differed, according to the types of available data for each region.

There are different ways to combine information from these three categories of data. The simplest of these methods is the use of majority opinion voting. If all three data sources are available, this method identifies a record as a fault when at least two of the three information sources confirm the occurrence of a fault in a specific cell and at a particular time. Another method is to use weighted voting. The basis of this method is the unequal accuracy of different data sources in fault detection. Therefore, assigning more weight to methods with higher precision is necessary. In this way, decisions are made about new data based on each method's ability to correctly identify faults in the network and consider the system's threshold.

As we mentioned earlier, all data sources may not exist simultaneously in this framework. In addition, according to conversations with field experts, who emphasize the necessity of checking all the reported faults from the performance support system and drive test sources, all the reported faults from these two sources should be checked. If both of these sources report the absence of a problem for a region and at a particular time, and there is a subscriber complaint for this region, this complaint is ignored. Otherwise, the common complaint is used to improve the accuracy of diagnosis of the fault's cause (Table 9).

**Table 9.** Checking different situations in combining information according to available data sources.

| The Number of Available Data Sources | Used Method |
| --- | --- |
| Only source number 1 reports an fault. Source number 1 and 2 report an fault | Check the problem |
| Source number 1 and 3 are not available | Trust first source |
| Source number 1, 2 and 3 are available | Trust first and second sources |

*3.16. Fault Identification*

Since the first and second data sources have two different views of the network and the events within the network, the type of detected fault from these two sources differ. Drive testing accurately identifies fault location and cause when a data source (performance support system) cannot respond appropriately. These two data sources are complementary to each other. When there are complaints from subscribers in an area, an expert is used to improve the accuracy of the solution provided.

As mentioned in the third section, it is necessary to examine the key performance indicators discussed in the second section for the short call scenario. Identifying the reason for blocking a call request using measurable parameters and features is not achieved in the drive test. Therefore, according to the findings of the second section, it is necessary to comment on the problems of the traffic sector and signaling sector for the desired area. Related indicators to request blocking in the signaling sector include congestion in the traffic and signaling channels, the success rate in accessing the traffic and signaling channels, and interruption in the signaling channel.

Among the drive test data, there were eight blocked call requests. The results of comparing these requests with the indicators of the busiest hours of the same day are shown in the Table 10.

**Table 10.** Comparison of drive test information and performance support system for blocked calls.

| ID | C/I | RxLe | RxQual | TA | SDCong | SDDrop | SDestab | TCHCong | TCHAssign |
|----|------|------|--------|----|--------|--------|---------|---------|-----------|
| 1 | 11.4 | −91 | 7 | 2 | 0.42 | 0.79 | 96.71 | 2.2 | 2.77 |
| 2 | 17.9 | −79 | 0 | 1 | 6.61 | 0.53 | 97.56 | 0 | 0.49 |
| 3 | 13.2 | −76 | 4 | 2 | 1.41 | 0.38 | 98.11 | 0 | 1.09 |
| 4 | 8.7 | −79 | 7 | 3 | 0 | 0.27 | 88.11 | 0 | 0.34 |
| 5 | 11.7 | −77 | 4 | 3 | 0 | 3.71 | 88.20 | 0 | 1.39 |
| 6 | 12.5 | −82 | 5 | 1 | 0.43 | 0.46 | 96.17 | 1.81 | 14.54 |
| 7 | 13.1 | −83 | 6 | 1 | 0 | 1.27 | 89.18 | 0.24 | 22.16 |
| 8 | 15.4 | −91 | 7 | 2 | 0 | 1.31 | 96.86 | 0 | 5.11 |

It is evident that records 1, 6, and 8 did not have signaling problems, and the reason for the call blocking problem was related to the indicators of the traffic department. Thus, records 2, 3, 4, and 5 were almost healthy in establishing traffic department calls, and there was no need to perform traffic checks on the blocked calls. It was possible to identify the fault by the method of determining the cause and using the performance support system data.

In this section, after introducing the third category of information, called subscriber complaints, we discussed the issue of combining different available information sources to identify the fault and also to analyze its cause. In this process, the main focus is on two sources of information, performance support systems and drive testing to identify the fault. A significant percentage of the complaints raised by subscribers is not a priority in solving network problems, and, from an economic pint of view, and even a technical point of view, are not cheap to resolve. Complaints from subscribers are only used to help the expert in identifying the cause of a fault that occurred in the system. Further investigations and analysis of subscribers' complaints to determine their importance in expressing the problem and helping to solve it are proposed as one suggestion for future work in this research. As mentioned before, the two information sources of drive test and performance support system data complement each other in fault detection and identify its cause. The combination of these two sources of information to identify the reason for the blocking of a call request in the drive test was investigated in this section, and how to find the cause of the fault in the field test, by using the method of identifying the fault caused in the data of the performance support system, was discussed.

## 4. Discussion and Conclusions

Troubleshooting in cellular networks is essential due to the nature of the networks' components, hardware, and software problems. Considering the competitive world among cellular network operators, the need for automatic fault detection and identification of the causes of faults, so as to restore the network to its normal mode, have become increasingly apparent. This article aimed to provide a framework for automatic fault detection and investigation of the cause of a fault to help the human resources in this field. This article was based on scientific principles and sought to solve problems in the industry, and its modeling was performed in consultation with experts in the field. Unlike other research conducted in this field, this research had different data sources, and by using the ability of the data to check the network mode from different perspectives, it identified faults with higher accuracy and exhibited greater ability to analyze the cause of fault. In addition, the proposed model had the most negligible dependence on experts in building the model and its initial assumptions. Therefore, it was immune to human faults and experts' taste.

Finally, in responding to the needs of experts, this model had a minor dependency on human resources and could continue to work without human resources' intervention. In this paper, the general framework of the activities was shown in "Fault! Reference source not found". The considered input sources were performance support system data, drive test data, and data related to customer complaints. Fault detection in the case of the first two categories of data, performance support system data and drive testing, was performed by methods fully explored in the second and third sections. Faults identified from the data sources were entered into the "combining data" section, along with potential subscriber complaints. In this part, faults were collected together with the knowledge extracted from the previous steps, and a decision was made for an area. Therefore, the input of the three parts, "performance support system," "drive test," and "subscriber complaints", was the available data from the three types of sources, and in the output port, the detected fault characteristics were entered into the information combination part. Finally, the detected fault was sent to the output and cause.

The performance support system is one of the most important sources of information in identifying network problems. The data of this system were examined in detail in the second section. The CCC quality criterion was used for fault detection. The records identified as faults by this criterion were entered into the next step of the algorithm to determine the cause of the fault. These data were divided into traffic and signaling data categories, and the related problems to each section were identified separately. Since the available data were only unlabeled raw data, the clustering algorithm method was used. By implementing different algorithms with different numbers of clusters, 5 clusters with a Silhouette coefficient of 0.4509 for traffic data and 6 clusters with a Silhouette coefficient of 0.503 were obtained for signaling data. Each of these clusters represented a specific cause for a fault in the network. Finally, different classification algorithms were applied to the labeled data through clustering to better evaluate the results. For traffic and signaling data, combining the effects of different classifiers through opinion voting had the best accuracy in test data. In fact, 90.38% accuracy was obtained for traffic data and 92.31% for signaling data, which was a significant improvement compared to the accuracy of other similar tasks. Drive test data were collected in three short, long and idle mode call scenarios. The short call identifies network problems in call setup, the long call identifies issues related to handover and call interruption, and, finally, the idle mode ascertains characteristics of the standard signal in the network. This research used performance support system data to solve the problems of blocked calls in short calls, long calls, and idle mode and used clustering algorithms to identify the cause of existing problems. For the accuracy of this method, in addition to the Silhouette coefficient, various classified algorithms were performed on the training and test data in the case of long call scenarios. In the best case, an accuracy of 96.86% was obtained with the dynamic neural network method. In addition, the time window was used to provide a framework for identifying points that needed handover, and its results were presented in the third section of the outsourcing review section. As mentioned in the first section, the number of studies conducted in this field has been minimal, and this study can be expanded to various areas. Examining subscriber complaint data in more detail, including identifying the importance of the reported problem for operators, is one of the essential activities to reduce the time spent by experts in this industry. In addition, according to the record that the subscriber registers in the system and the explanations that they provide to the subscriber complaints center, it is possible to identify the fault type and analyze its cause.

**Author Contributions:** Conceptualization, A.K.S.; Software, A.K.S., S.R. and A.J.; Formal analysis, S.R., A.J., F.M., W.Z. and D.W.; Investigation, F.M. and W.Z.; Resources, A.K.S., F.M. and D.W.; Data curation, A.K.S., S.R. and A.J.; Writing—original draft, A.K.S., S.R. and A.J.; Writing—review & editing, A.J.; Visualization, A.J.; Supervision, A.J. and W.Z.; Project administration, A.J. All authors have read and agreed to the published version of the manuscript.

**Funding:** This work was supported in part by the Fundamental Research Funds for the Central Universities under Grant No. HIT.OCEF.2021007, the Shenzhen Science and Technology Research and Development Foundation under Grant No.JCYJ20190806143418198, the National Key Research and Development Program of China under Grant No. 2020YFB1406902, the Key-Area Research and Development Program of Guangdong Province under Grant No. 2020B0101360001, the Guangdong Provincial Key Laboratory of Novel Security Intelligence Technologies under Grant No. 2022B1212010005. Professor Weizhe Zhang is the corresponding author.

**Data Availability Statement:** The data that support the findings of this study are available from the corresponding author, upon reasonable request.

**Conflicts of Interest:** The authors declare no conflict of interest.

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
