# Peer review of "Automatic Fault Detection and Diagnosis in Cellular Networks and Beyond 5G: Intelligent Network Management"

_algorithms, doi:10.3390/a15110432_

Round 1
Reviewer 1 Report
The performance support system is one of the most important sources of information in identifying network problems. The data of this system have been examined in detail in the second section. CCC quality criterion was used for error detection. The records identified as errors by this criterion are entered into the next step of the algorithm to determine the cause of the error.
The authors are encouraged to revise the current manuscript because some important points must be clarified or fixed.
1. There are several grammatical issues throughout the paper. It would be better if the author rechecks the paper's English writing for possible grammatical errors and typos. For example, “is” in the first line in the introduction “Sustainable Computing and Cloud computing is…” moreover, some citations were added after the dot, please check the whole of the manuscript to solve such similar issues.
2. The authors are recommended to rewrite the abstract and highlight the novelty and findings of this study precisely.
3. The authors are suggested to describe the importance and contribution of this study in the Introduction more specifically.
4. It is recommended to provide a sufficient description of the shortcomings of the prior studies in the research background and literature review section.
5. The authors are recommended to check Figures 2-4 and provide the citation for it. The quality of Fig 6 should be improved and also its description is not clear. Some items such as X, Y, etc. should be determined.
6. The authors are recommended to add a paragraph(s) and explain the contents of Table 1 clearly.
7. It is recommended to determine the formulation of this study (single/multi-objective, response time, and effectiveness).
8. The authors are recommended to provide a schematic diagram of the proposed method. Moreover, please determine which section of the proposed method section is the novelty of this study.
9. It is recommended to insert the subsections of the proposed method which does not belong to the novelty of this study into a preliminary section. For example “various classified algorithms on performance support system data “ Please rewrite the proposed methods section by considering this comment.
10. The detail of regression such as type, arguments, etc. is not clear.
11. The authors are recommended to show which table can support the following sentence. “The results show that using the regression for categorizing the CCC to manage the market leads to an improvement in Equation 2-2: Calculating the qualitative measure of network performance.”
12. Please check the second item in the legend of Fig. 7. Moreover, in Fig. 13 the first box “Architecture of working with drive test data” cannot show the architecture and more details are needed.
Minor Comments:
- What TCH assignments algorithm stands for ...
Regarding the objectives:
How the techniques affect Silhouette coefficient requirements was stated but not discussed in the subsequent Sections. Studying the effect of certain metrics require implementation of each technique and thorough evaluation of their impact on that metric.
By lowering overhead and the number of cluster head changes, Artificial Intelligence Transition and TCH can be optimized by minimizing or maximizing the objective function, but this is not mentioned later.
Regarding the Related work:
You must compare the proposed work with current literature to see the differences and identify the paper's key contribution to the field.
The comparison table in this paper has a deep depth, making it suitable to use in this paper.
What is the condition of the vehicles in this research? How does the algorithm consider and use queues for data forwarding? This will affect the quality of service (Using self-organizing networks (SON) /QoE)? An explanation of Self-healing networks simulation area and Artificial Intelligence matrix dataset is needed.
The purpose of the Artificial Intelligence chain property and transition matrix in 5G is to predict future events. Is it possible to categorize vehicles based on two separate priority queues based on Artificial Intelligence? If you add a clustering mechanism, you can more accurately predict
Algorithm 1: what is Procedures for making a call and (X(t)) stands for?
Algorithm 1: Choosing a , randomly generating the values x1, x2, x3, .., xn according to the distribution of the values doesn't mean you've found the optimal statistical setup You should consider more factors in the selection criteria. I would like to see more explanation on why this is you chosen metric.
the standard mac protocol already has a priority messaging mechanism that consider high priority messages in its dissemination, is that mean you are using the same one or proposing a new one?
Author Response
The Author's Notes to the Reviewer have been attached.

Reviewer 2 Report
It is not clear from the whole large article - what did the authors do in the end? Multiple reasoning about clustering, but what's the bottom line? Figure 18 is quite general.
Page 36: "Therefore, paying attention to subscribers' complaints is one of the critical issues in identifying the error cause."
The authors mention that when analyzing errors, it is necessary to take into account users reports. How it's done?
Author Response

(The authors gave the same response as above.)

Round 2
Reviewer 1 Report
All issues have been successfully addressed by authors.
Author Response
Thank you for your comments.
Reviewer 2 Report
Fig. 18 typo: Syetems
pp. 37-39 contain absolutely qualitative reasoning. What was the automation of the investigation of errors is not clear.
Author Response
Fig. 18 typo: Syetems
Revised.
pp. 37-39 contain absolutely qualitative reasoning. What was the automation of the investigation of errors is not clear.
This is Error and Fault Detection in Cellular Network Management. We have calculated based on Simulation and real data (Hamrahe Aval)
Round 3
Reviewer 2 Report
Fig. 11 is not readable
Fig 4 and Fig. 13 - are the same?
Table 7 - clarification needed. Images 2 and 3 are almost the same. But in one case everything is fine, in the other - problems. Why?
Page 38: "It is evident that records 1, 6, and 8 do not have signaling problems"
To whom is it obvious and why is it obvious? Which one(s) of the 10 table parameters indicate this?
Round 4
Reviewer 2 Report
All my comments have been taken into account.